# Determination of reference intervals for knee motor functions specific to patients undergoing knee arthroplasty

**Hideyuki Ito[1], Kiyoshi Ichihara**[2]*, **Kotaro Tamari**[3], **Tetsuya Amano**[4], **Shigeharu Tanaka**[5], **Shigehiro Uchida**[6], **Shinya Morikawa**[7]

**1** Department of Rehabilitation, Faculty of Wakayama Health Care Sciences, Takarazuka University of Medical and Health Care, Wakayama, Japan, **2** Faculty of Health Sciences, Yamaguchi University Graduate School of Medicine, Ube, Japan, **3** Department of Physical Therapy, Faculty of Health and Medical Science Teikyo Heisei University, Tokyo, Japan, **4** Department of Physical Therapy, Faculty of Health and Medical Sciences, Tokoha University, Hamamatsu, Japan, **5** Division of Physical Therapy, School of Rehabilitation, Faculty of Health and Social Services, Kanagawa University of Human Services, Yokosuka, Japan, **6** Department of Rehabilitation, Faculty of Rehabilitation, Hiroshima International University, Higashihiroshima, Japan, **7** Department of Rehabilitation, Hoshasen Daiichi Hospital, Imabari, Japan

* ichihara@yamaguchi-u.ac.jp

**Data Availability Statement:** All relevant data are within the manuscript and its Supporting information files.

## Abstract

### Background

In patients with knee osteoarthritis (KOA) undergoing knee arthroplasty (KA), lower-limb motor function tests are commonly measured during peri-surgical rehabilitation. To clarify their sources of variation and determine reference intervals (RIs), a multicenter study was performed in Japan.

### Methods

We enrolled 545 KOA patients (127 men; 418 women; mean age 74.2 years) who underwent KA and followed a normal recovery course. The surgical modes included total KA (TKA), minimally invasive TKA (MIS-TKA), and unicompartmental KA (UKA). Motor functions measured twice before and two weeks after surgery included timed up-and-go (TUG), maximum walking speed (MWS), extensor and flexor muscle strength (MS), and knee range of motion (ROM). Multiple regression analysis was performed to evaluate their sources of variation including sex, age, BMI, and surgical mode. Magnitude of between-subgroup differences was expressed as SD ratio (SDR) based on 3-level nested ANOVA. SDR$\geq$0.4 was set as the threshold for requiring RIs specific for each subgroup.

### Results

Before surgery, age-related changes exceeding the threshold were observed for TUG and MWS. Between-sex difference was noted for extensor and flexor MS, but extension and flexion ROMs were not influenced by sex or age. After surgery, in addition to similar influences of sex and age on test results, surgical modes of UKA and MIS-TKA generally had a favorable influence on MWS, extensor MS, and flexion ROM. All motor function test results

**Funding:** The authors received no specific funding for this work.

**Competing interests:** The authors have declared that no competing interests exist.

showed a variable degree of skewness in distribution, and thus RIs were basically derived by the parametric method after Gaussian transformation of test results.

## Conclusions

This is the first study to determine RIs for knee motor functions specific to KOA patients after careful consideration of their sources of variation and distribution shapes. These RIs facilitate objective implementation of peri-surgical rehabilitation and allow detection of patients who deviate from the normal course of recovery.

## Introduction

Yoshimura et al. [29] reported in a large cohort study across Japan that the prevalence of knee osteoarthritis (KOA) is as high as 42.6% in men and 62.4% in women over 40 years old. Their report also disclosed that among the 25.3 million patients with KOA, 8 million have symptomatic disease, which poses a large socio-economic problem in Japan [1]. KOA features degeneration and attrition of joint structure that result in ossification of the cartilage and surrounding tissue. This restricts knee range of movement causing pain and gait disturbance. Conservative therapy, such as exercise, ultrasonic therapy, electrotherapy, and orthosis therapy, is the primary treatment choice. However, with the development of multiple osteophytes, sclerosis, and narrowing of joint space, knee arthroplasty eventually becomes necessary. Either unicompartmental knee arthroplasty (UKA) or total knee arthroplasty (TKA) has been the therapeutic regimen of choice. The former is a less invasive technique that only replaces a single granular area and preserves the anterior and posterior cruciate ligaments [2, 3]. Recently, as an alternative to conventional TKA (C-TKA), a minimally invasive TKA surgery (MIS-TKA) has become popular, which features a shorter skin incision than that with C-TKA.

In the clinical management of KOA patients undergoing knee arthroplasty, physical therapists take an important role in providing peri-surgical rehabilitation in close collaboration with orthopedic surgeons. The scheme of rehabilitation according to a common clinical pathway for knee arthroplasty is shown in Fig 1. The functional status of patients is usually assessed by performing knee motor function tests of a safe variety, such as the timed-up-and-go (TUG) test, maximum walking speed (MWS), knee muscle strength, and knee range of motion (ROM) [4]. These test results are often referred to in deciding a patient's discharge date. To read the results objectively, it is essential to understand their sources of variation (factors that may alter test results without pathological conditions) and to have a reliable reference interval (RI) for each parameter. To the best of our knowledge, there have been no published studies that evaluated motor function tests of patients under the same situation despite the high demand for knee replacement surgery.

Although several reports have presented so-called "healthy" normative values for knee motor function tests [5–9], these normative values were reported only in the form of mean ± standard deviation (SD) of test results without consideration of the shapes of their distribution. Therefore, a RI that corresponds to the true central 95% range of test results is not certain for practical use. In recent years, in the field of clinical chemistry and laboratory medicine, RIs are determined as a central 95% range of healthy values either nonparametrically as a 2.5 to 97.5 percentile range of values or parametrically in three steps: (1) transformation of the distribution of test results into Gaussian shape, (2) calculation of the central 95% range as mean ± 1.96 SD, and (3) reverse-transformation of the range back to the original scale [10, 11].

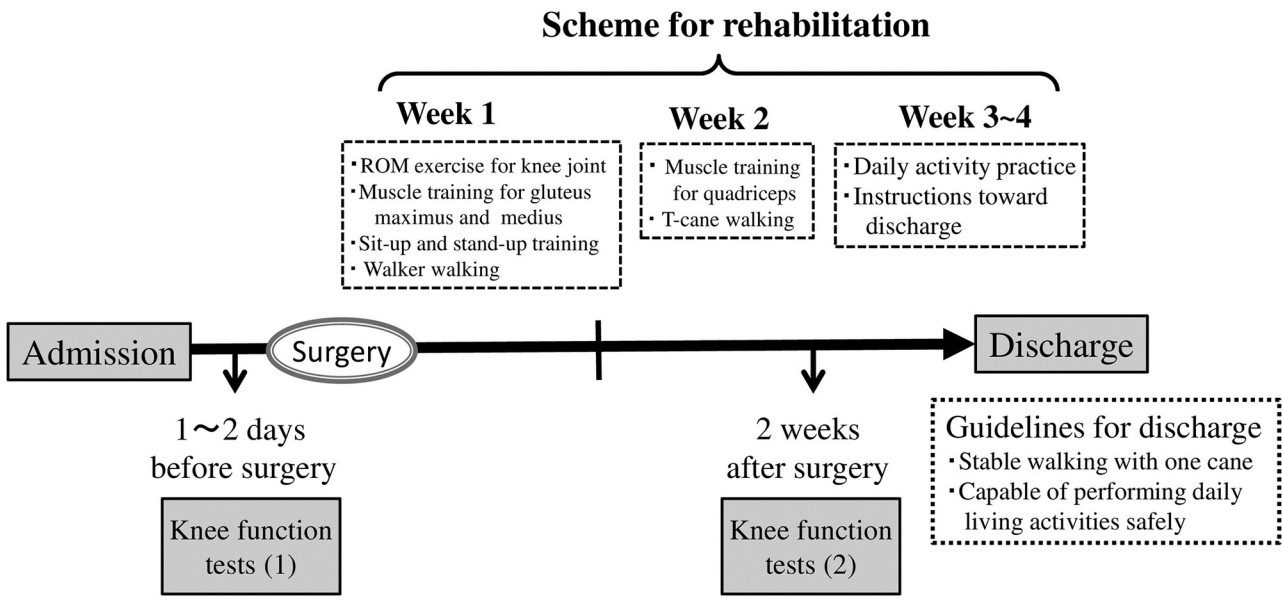

**Fig 1. Clinical pathway for knee replacement surgery.**

Another problem with using the reported healthy normative values was that they were generally not matched to KOA patients, who have compromised knee motor functions due to the interference of the operative-side knee or the presence of KOA on the non-operative side as well.

On the basis of this background, we conducted a multicenter study to evaluate sources of variation (SVs) of knee motor functions and to establish well-stratified RIs in consideration of the SVs for KOA patients undergoing knee arthroplasty. Age matched healthy subjects were also recruited to ascertain the need for disease specific RIs. We aimed to use these quantitative information for objective implementation of peri-surgical rehabilitation and for screening of patients who deviate from the normal course of recovery.

## Subjects and methods

### Study subjects

**Patients.** In total, 624 KOA patients undergoing elective knee arthroplasty were recruited between July 2013 and February 2018 from 13 institutions across Western Japan that specialize in knee arthroplasty. The inclusion criteria were 1) diagnosed as having osteoarthritis and 2) having a clinical indication for TKA or UKA. The exclusion criteria were 1) presence of motor paralysis or other neurological dysfunctions such as stroke, lumbar disc herniation, spinal canal stenosis, or peripheral neuropathy due to diabetes mellitus unrelated to KOA, 2) obvious motion pain or restricted joint movements in locations other than the knee, which would hinder walking or the sit-to-stand motion, 3) cognitive impairment, 4) missing test results for motor functions tests as planned, 5) failure to be discharged because of some deviation from the clinical pathway, such as walking instability, persistent inflammation, surgery-related fracture, deep venous thrombosis, or post-surgical infection. After exclusion of 38 patients with missing motor function test results and 41 patients with delayed discharge, 545 patients were considered eligible for the subsequent analyses.

The breakdown of cases by surgical mode was C-TKA, 99 cases; MIS-TKA, 342 cases; and UKA, 104 cases, where MIS-TKA was defined as a surgical mode involving a skin incision width of <5 cm [12]. The surgical mode of UKA features less bone resection and less injury to the quadricep muscles compared to TKA [13, 14].

The patients were managed following a standardized clinical pathway based on a rehabilitation program implemented in common in the 13 institutions (Fig 1). The planned date of discharge was set at 3–4 weeks after the surgery. The common program was composed of the following for the first post-surgical week: exercise for knee joint ROM, muscle training for gluteus maximus and gluteus medius, sit-up and stand-up training; for the second week: muscle training for quadriceps and T-cane walking; and for the third week: staircase training, daily activity practice, and instructions toward discharge. These motor function tests were performed twice, just before and two weeks after the surgery. The general guideline of allowing discharge at 3–4 weeks after knee arthroplasty was based on attainment of stable walking with one cane and the capability of performing daily living activities safely.

**Healthy volunteers.**   As described in the Introduction, we recruited 120 apparently healthy volunteers with comparable ages to make a contrast of their knee motor functions with those of KOA patients undergoing knee arthroplasty. They comprised 36 men and 84 women mostly from elderly clubs registered in a municipal health promotion office. The inclusion criteria adopted from Bohannon [15] were 1) self-awareness of physical and mental health, 2) self-efficacy for leading independent daily life, 3) not currently under medical care for motor or neurological diseases or cardiac diseases, 4) no previous history of knee or hip join replacement surgery, and 5) capable of walking 30 meters without any assistance or device.

## Ethical considerations, explanation, and consent

This study was conducted strictly following the Declaration of Helsinki and "Ethical Guidelines on Clinical Studies". The objective and protocol of this study were evaluated and approved by the Tokoha University Ethics Committee (approval no.: 2018-501H). We provided a written explanation of this study in language that was understandable to the participants and obtained written consent from each subject after confirming their understanding and acceptance.

## Research design

This study corresponded to a prospective cohort study for establishing RIs of knee motor functions specific to KOA patients.

## Measurements

**Basic physical and medical attributes.**   We recorded sex, age, body mass index (BMI), Kellgren-Lawrence classification [16, 17] for the severity of knee joint degeneration (K-L classification), affected sides (unilateral or bilateral), and surgical mode (C-TKA, MIS-TKA, or UKA). Regular exercise was defined as exercise performed a minimum of twice weekly for 30 minutes or more.

**Motor functions.**   The measurements of the knee motor functions were performed according to the following procedures.

*Timed up-and-go*. The examinee was asked to sit comfortably on an armless chair with seat-height set between 40 and 45-cm high with their back leaning against the backrest and both hands on the thighs.

On the vocal "go" signal, the examinee got up from the chair, walked 3 meters, turned around, and returned to sit on the chair. The time required for the entire movement was

measured by stopwatch. This test was done in two ways: at a comfortable speed (TUGcom) of usual walking [18] and at the patient's maximum walking speed (TUGmax) [19].

*Maximum walking speed.* The test distance for MWS measurement was set to 5 meters. The examinee was allowed 3 meters each for acceleration and deceleration before and after the test distance. The examinee was asked to safely walk as fast as possible without running as recommended by Dobson [4] and Fransen [20] for increased reproducibility. On the vocal "go" signal, the examinee started to walk from the line set 3 meters before the test distance. A stopwatch was used to record walking time on reaching both ends of the test distance. The MWS (m/sec) was calculated as 5 m ÷ walking time (sec) [9].

*Knee extensor muscle strength.* The measurement was performed by use of a hand-held dynamometer (HHD) (µTas F-1; Anima Corp., Tokyo, Japan) according to the following procedures (Fig 2). The examinee was placed in a sitting position with both arms crossed in front of the chest. The examiner stood in front of the patient's lower leg being tested and attached the HHD-sensor belt around the front of the ankle with its belt fixed to the stool leg (Fig 2B). The examinee was then requested to apply maximum force to extend the lower leg against the ankle belt. The isometric force/torque (Nm) was recorded as a product of the HHD output (N) and lower leg length (m). Muscle strength (Nm/kg) was expressed as torque divided by body weight (kg). The lower leg length was measured as a distance between the distal end of the lower leg (where the center of the HHD sensor was placed) and the lateral cleft of the knee joint [21–23]. The muscle strength was measured twice, and the average value was used because variability of the measurement was higher in elderly women as reported by Katoh et al [24].

*Knee flexor muscle strength.* The strength of the knee flexor muscles was measured in a similar manner. The only difference was that the HHD sensor was placed around the back side of the ankle with its belt fixed to the examiner's leg (Fig 2D). The examinee was asked to apply maximum force to flex the knee to an acute angle against the belt. The strength was recorded as muscle strength of the knee flexor [21–23].

*Measurement of knee flexion and extension range of motion.* The ROM of the knee joint during flexion and extension was measured by a joint angle meter according to the guideline issued jointly by the Japan Orthopedic Surgery Society and the Japan Rehabilitation Medical Society [25]. The angle of the long axes of the tibia (on a line connecting the fibula head and lateral malleolus) and the femur (on a line connecting the greater trochanter and lateral epicondyle of the femur) was measured in units of 5 degrees.

## Statistical methods

**Sources of variation of motor function parameters.** To identify factors associated with motor function parameters, multiple regression analysis (MRA) was performed. Values of each of the motor function tests were set as a dependent variable. The following parameters that we could obtain from all subjects were evaluated as explanatory variables: sex, age, BMI, habit of regular exercise (binary: no = 0, yes = 1), and K-L classification (ranking scale of 0 to 4). In the analysis of motor function tests performed after the surgery, the additional explanatory variable of surgical mode (dummy variables were created for MIS-TKA and UKA by setting C-TKA as the reference category) was introduced in the regression model.

All explanatory variables were fixed for all objective variables for ease of comparability across motor function parameters. In addition to expressing the statistical significance of each standardized partial regression coefficient (rp) as a P value, we marked its effect size (practical significance) by use of a threshold of $|rp| \geq 0.2$, which corresponds to a midpoint effect size of Cohen's d of small (0.1) and medium (0.3) for correlation coefficients [26].

A

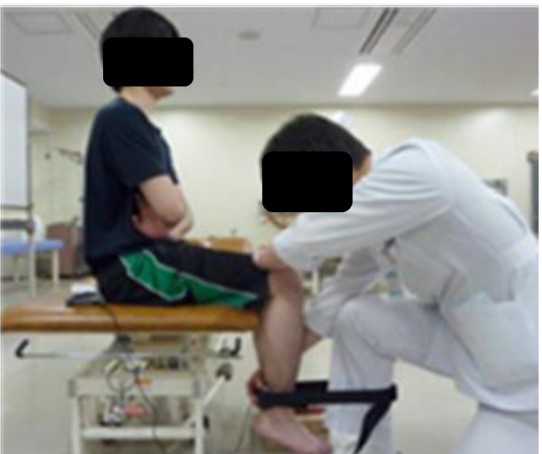

hand held dynamometer

B

extensor muscle strength

C

lower leg length

D

flexor muscle strength

**Fig 2. Measurement of knee extensor/flexor muscle strength.** The measurement was performed by use of a hand-held dynamometer (HHD) [A]. For measuring extension muscle strength [B], the examinee sitting on chair with the knee joint bent at 90-degree angle was asked to extend the lower leg at maximum force against the black belt around the ankle, which was fixed behind to the chair with the HHD attached to its front. The length of the lower leg (Len in meter) was measured [C] as a distance between the lateral cleft of the knee joint and the center of the HHD sensor at the ankle. The muscle strength in N·m was computed as a product of HHD output (N) and leg length (m). Similarly, the flexion muscle strength was measured [D] by forcibly flexing the lower leg with the belt fixed to the leg of the examiner and the HHD attached at the back side of the belt.

For motor function parameters before the surgery, three-level nested ANOVA was performed by setting the three factors to sex, age, and BMI, whereas for the parameters after the surgery, the ANOVA was performed by setting the factors to sex, age, and surgical mode. The choice of these three factors was determined based on the MRA results (see Results). In the analyses, age and BMI were converted into a ranking scale by partitioning values at 70 and 80 years for age (3 categories), and at 18.5, 25, and 30 kg/m$^2$ for BMI (4 categories according to

the WHO criteria of <18.5: underweight; 18.5≤–<25: normal range; 25≤–<30: pre-obese; and 30≤–<35: obese class I).

The analysis provided the magnitude of each source of variation in terms of the SD: SD for sex (SDsex), SD for age (SDage), SD for BMI (SD$_{BMI}$), and SD for surgical mode (SDsur) together with the SD for between-individual variations (SDindiv). The relative magnitude of each SD was expressed by determining its ratio to SDindiv. We designated it as SD ratio or SDR. Namely SDRsex = SDsex/SDindiv; SDRage = SDage/SDindiv; SDR$_{BMI}$ = SD$_{BMI}$/SDindiv; and SDRsur = SDsur/SDindiv. The threshold value for SDR at which we chose to partition test results by the factor was set to 0.4 as reported elsewhere [10, 11].

Before performing MRA and ANOVA, the values of TUG were converted logarithmically, and the values of extensor/flexor muscle strength were transformed to the power of 0.5 (square root transformation) to reduce the skewness of their distributions.

**Method for statistical derivation of RIs.** RIs for motor function parameters were basically derived by use of a parametric method that features power transformation of a value (test result) to make its distribution Gaussian by use of the following modified Box-Cox formula [27]:

$$X = \frac{(x-a)^p - 1}{p}$$

where $X$ represents the transformed value of test result $x$, and $p$ and $a$ represent power and an origin of transformation to be estimated by maximum likelihood method, respectively.

The central 95% interval under the transformed scale (LL$^T$, UL$^T$) can be calculated by use of the mean and SD of the transformed test results (m$^T$, SD$^T$) as

$$LL^T = m^T - 1.96\,SD^T$$

$$UL^T = m^T + 1.96\,SD^T$$

Then, the lower and upper limits under the original scale (LL, UL) can be calculated by reverse transformation using the following formulae:

$$LL = (p \times LL^T + 1)^{1/p} + a$$

$$UL = (p \times UL^T + 1)^{1/p} + a$$

The success of the Gaussian transformation was confirmed by the linearity of the cumulative frequency curve on the probability plot as well as by the Kolmogorov-Smirnov test of normality.

When test results of a given parameter failed in the Gaussian transformation by the Box-Cox method or when test results took discrete values, we applied a nonparametric method for determining the central 95% range of test results after sorting the values and took the range of 2.5 and 97.5 percentiles as the limits of the RI.

## Results

### Profiles of KOA patients

The patients with KOA comprised 127 men and 418 women with respective mean ages ± SD of 74.4 ± 8.0 and 74.2 ± 7.6 years and BMIs of 25.3 ± 3.6 and 25.3 ± 3.6 kg/m$^2$ as summarized in Table 1. The habit of regular exercise was self-reported as "yes" by 176 and "no" by 369 patients. The affected knee joints were unilateral in 205 and bilateral in 340 patients. The

**Table 1. Demography of patients and healthy volunteers.**

| | KOA patients | | | healthy volunteers | P-value |
|---|---|---|---|---|---|
| N | 545 | | | 120 | |
| Sex (n) | men:127 women: 418 | | | men:36 women: 84 | 0.12 |
| Age (year) | 74.2 ± 7.7 | | | 71.3 ± 5.9 | 0.0001 |
| BMI (kg/m$^2$) | 25.3 ± 3.7 | | | 22.5 ± 2.9 | 0.0000 |
| Exercise (n) | Yes:176 no:369 | | | Yes:99 no:21 | 0.0000 |
| K-L class (n) | Grade 2:29 | Grade 3:252 | Grade 4:264 | | |
| Surgical mode (n) | C-TKA: 99 | MIS-TKA: 342 | UKA: 104 | | |
| Affected knee (n) | unilateral:205 bilateral:340 | | | | |
| Days of admission | 25.7 ± 6.3 | | | | |

severity of the knee joint deformity by K-L classification was Grade 2, 3, and 4 in 29, 252, and 264 patients, respectively. The mode of surgery was C-TKA in 99 patients, MIS-TKA in 342, and UKA in 104.

## Demography of healthy controls

The healthy controls comprised 36 men and 84 women with respective mean ages ± SD of 72.3 ± 5.9 and 70.9 ± 6.0 years and BMIs of 23.2 ± 2.4 and 22.1 ± 3.1 kg/m$^2$. The habit of regular exercise was self-reported as "yes" in 99 and "no" in 21 subjects.

Differences between healthy subjects and KOA patients were tested by Mann-Whitney test for numerical variables and chi-square tests for categorical variables. Statistically significant differences were found for higher age and BMI, and less frequent habit of exercise observed among KOA patients.

## Sources of variation of motor functions among osteoarthritis patients

Regarding the motor function tests before the surgery, as shown in the upper half of Table 2, MRA revealed that age was a significant source of variation with |rp|>0.2 for TUGcom, TUGmax, and MWS with respective rp values of 0.32, 0.34, and −0.31. Sex was also significant for extensor and flexor muscle strength of the knee. In general, there were greater between-sex differences in extensor muscle (rp = −0.32) than in flexor muscle (rp = −0.24) strength. None of the factors were associated with knee extension ROM, whereas BMI was negatively associated with knee flexion ROM (rp = −0.22), i.e., the higher the BMI, the smaller was the flexion ROM.

For the motor function tests after surgery, as shown in the lower half of Table 2, age was positively associated with TUGcom and TUGmax (rp = 0.35 and 0.37) and negatively associated with MWS (rp = −0.34). The surgical mode of UKA was negatively associated with TUGcom and TUGmax (rp = −0.24 and −0.24) and positively associated with MWS (rp = 0.31), i.e., the surgical mode of UKA resulted in shorter TUG and faster MWS compared to C-TKA (the reference category). The surgical mode of MIS-TKA was also positively associated with MWS (rp = 0.22) compared to C-TKA, i.e., MWS was faster with MIS-TKA than with C-TKA. Furthermore, the women showed weaker extensor and flexor muscle strength (rp = −0.29 and −0.24), and UKA had a positive influence on the extensor muscle strength (rp = 0.22).

None of the factors were associated with the ROM of knee extension. For the ROM of knee flexion, however, the surgical mode of UKA showed a larger ROM compared to C-TKA, with a rp of 0.26. These findings revealed by MRA can be confirmed graphically as follows.

**Table 2. Multiple regression analyses for sources of variation of motor function parameters.**

| Before surgery | R | Sex | | Age | | BMI | | Exercise | | K-L class | | Bilateral | | | | | |
|---|---|---|---|---|---|---|---|---|---|---|---|---|---|---|---|---|---|
| TUG comfort | 0.402 | 0.11 | * | **0.32** | *** | -0.01 | | -0.11 | * | 0.14 | ** | 0.01 | | | | | |
| TUG maximum | 0.431 | 0.15 | *** | **0.34** | *** | 0.03 | | -0.08 | | 0.15 | ** | 0.01 | | | | | |
| Maximum walking speed | 0.436 | -0.16 | *** | **-0.31** | *** | 0.01 | | 0.12 | * | -0.17 | *** | -0.04 | | | | | |
| Extensor muscle strength | 0.355 | **-0.32** | *** | -0.12 | | -0.06 | | 0.06 | | -0.01 | | 0.03 | | | | | |
| Flexor muscle strength | 0.302 | **-0.24** | *** | -0.10 | | -0.11 | | 0.01 | | -0.07 | | -0.04 | | | | | |
| Extension ROM | 0.198 | -0.03 | | 0.05 | | -0.08 | | 0.05 | | -0.15 | * | 0.04 | | | | | |
| Flexion ROM | 0.296 | -0.11 | * | -0.02 | ** | **-0.22** | *** | 0.05 | | -0.09 | | -0.01 | | | | | |
| **After surgery** | **R** | **Sex** | | **Age** | | **BMI** | | **Exercise** | | **K-L class** | | **Bilateral** | | **MIS-TKA** | | **UKA** | |
| TUG comfort | 0.437 | 0.08 | | **0.35** | *** | 0.01 | | -0.04 | | 0.07 | | 0.04 | | -0.16 | * | **-0.24** | *** |
| TUG maximum | 0.468 | 0.12 | ** | **0.37** | *** | 0.00 | | -0.05 | | 0.10 | | 0.03 | | -0.15 | * | **-0.24** | *** |
| Maximum walking speed | 0.484 | -0.16 | ** | **-0.34** | *** | -0.02 | | 0.08 | | -0.07 | | -0.03 | | **0.22** | *** | **0.31** | *** |
| Extensor muscle strength | 0.367 | **-0.29** | *** | -0.10 | | -0.12 | ** | 0.00 | | 0.02 | | 0.06 | | 0.19 | ** | **0.22** | ** |
| Flexor muscle strength | 0.353 | **-0.24** | *** | -0.19 | *** | -0.15 | ** | -0.06 | | -0.03 | | -0.01 | | 0.15 | * | 0.10 | |
| Extension ROM | 0.205 | 0.11 | * | -0.04 | | -0.09 | | 0.00 | | 0.08 | | -0.02 | | 0.09 | | 0.11 | |
| Flexion ROM | 0.242 | -0.07 | | -0.08 | | -0.04 | | -0.01 | | 0.01 | | 0.01 | | 0.12 | | **0.26** | *** |

R = multiple correlation coefficient; Exercise (binary) indicating habit of regular exercise; K-L class (ordinal) from 2 to 4; Bilateral (binary) indicating bilateral involvement of osteoarthritis of the knees.

Multiple regression analysis (MRA) was performed by setting each motor function parameter as a dependent variable and a fixed list of explanatory variables: Sex, age, BMI, exercise, K-L class, bilateral. Listed are standardized partial regression coefficients ($r_p$), which take values between -1.0 and 1.0. The values above the effect size of $|r_p| \geq 0.2$ are highlighted by bold letter with light orange background. Statistical significance is also indicated as

* = $P < 0.01$;

** = $P < 0.001$, and

*** = $P < 0.0001$.

In Fig 3, the measured values of TUGcom (Fig 3A) and TUGmax (Fig 3B) were compared among healthy controls and patients before and after surgery subgrouped by age ($\leq$69, 70–79, $\geq$80 years) and surgical mode. The prolongation of TUG with age is obvious across all groups. TUG was increased in ascending order of UKA, MIS-TKA, and C-TKA. The figure also shows how much difference in TUG occurred in KOA patients before and after the surgery compared to the healthy elderly of comparable ages.

A similar comparison was made for MWS in Fig 3C. Both the healthy controls and the patients before surgery showed an age-related decrease in MWS. The reduction in MWS in the patients is prominent regardless of age compared to the healthy controls.

MWS after surgery was further subgrouped by surgical mode. Both age and surgical mode were associated with the levels of MWS both by MRA and ANOVA. However, surgical mode was relatively more intense ($SDR_{age} = 0.363$ vs $SDR_{sur} = 0.442$). Therefore, in this figure, postsurgical MWS was partitioned by the mode of surgery.

In Fig 4A and 4B, extensor and flexor muscle strengths were respectively compared after partitioning by sex between three groups: healthy controls and patients in presurgical and postsurgical states. Extensor muscle strength postoperatively is reduced in females.

In Fig 4C and 4D, extension and flexion ROM of the knee was respectively compared between healthy controls and KOA patients before and after surgery. The values of extension and flexor ROM of the patients were greatly decreased before the surgery compared to those of the healthy controls. However, after surgery, the values of extension ROM improved appreciably, whereas the values of flexion ROM decreased further after the surgery.

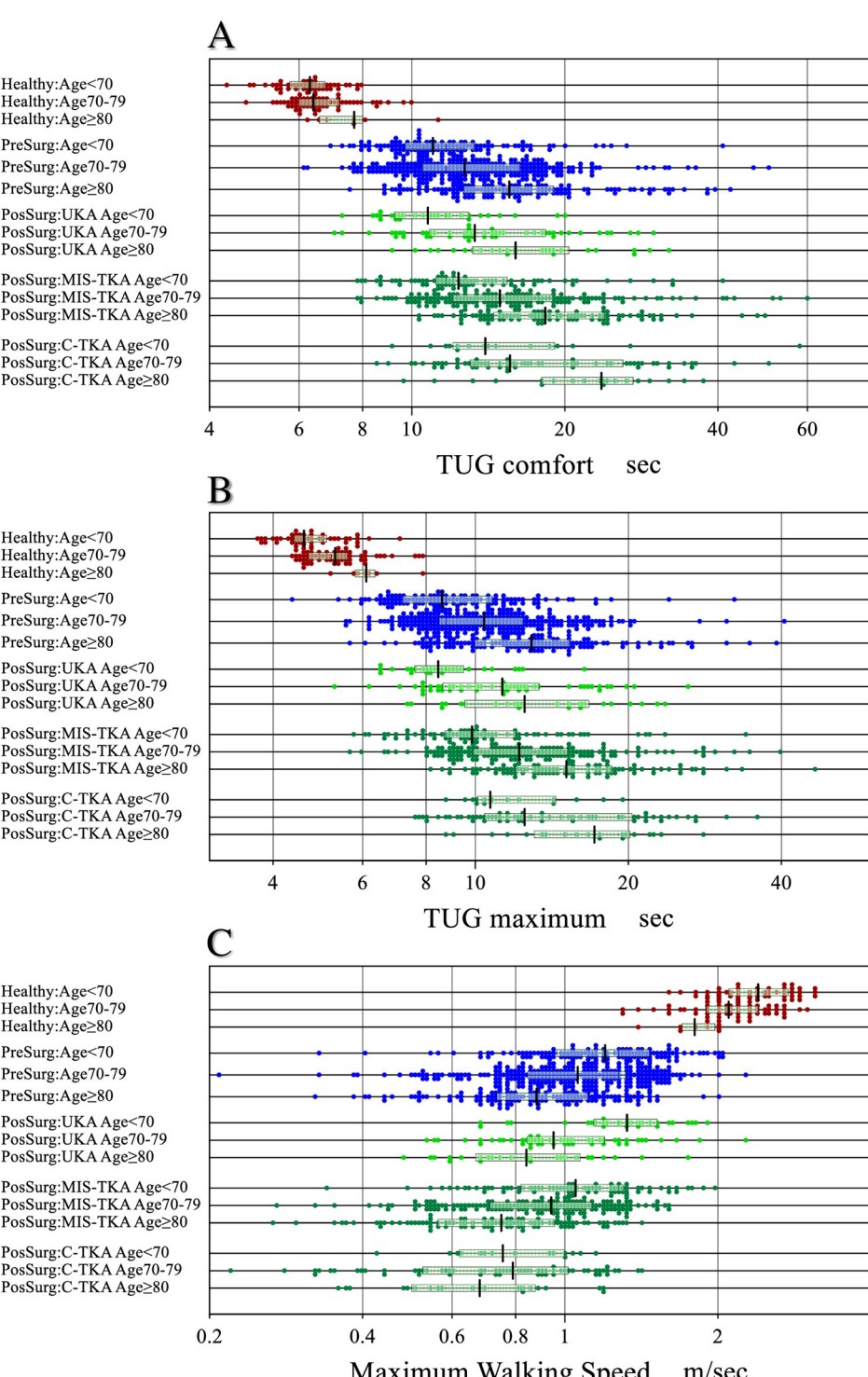

**Fig 3. Comparison of TUGs and MWS partitioned by age and surgical mode.** Measured results of timed up and go test (TUG) at comfortable (A) or at maximum (B) walking speed in seconds, and maximum walking speed (MWS) (C) were compared between three groups: Healthy controls and patients before and after surgery. Because of prominent age-related changes in the values, they were partitioned by age at 70 and 80 years, and for postsurgical values, also by surgical modes. The x-axis of TUG was transformed logarithmically and that of MWS was transformed to the power of 0.8 to make their distributions near Gaussian. The box and center line respectively represent the mid-50% range and median of each subgroup.

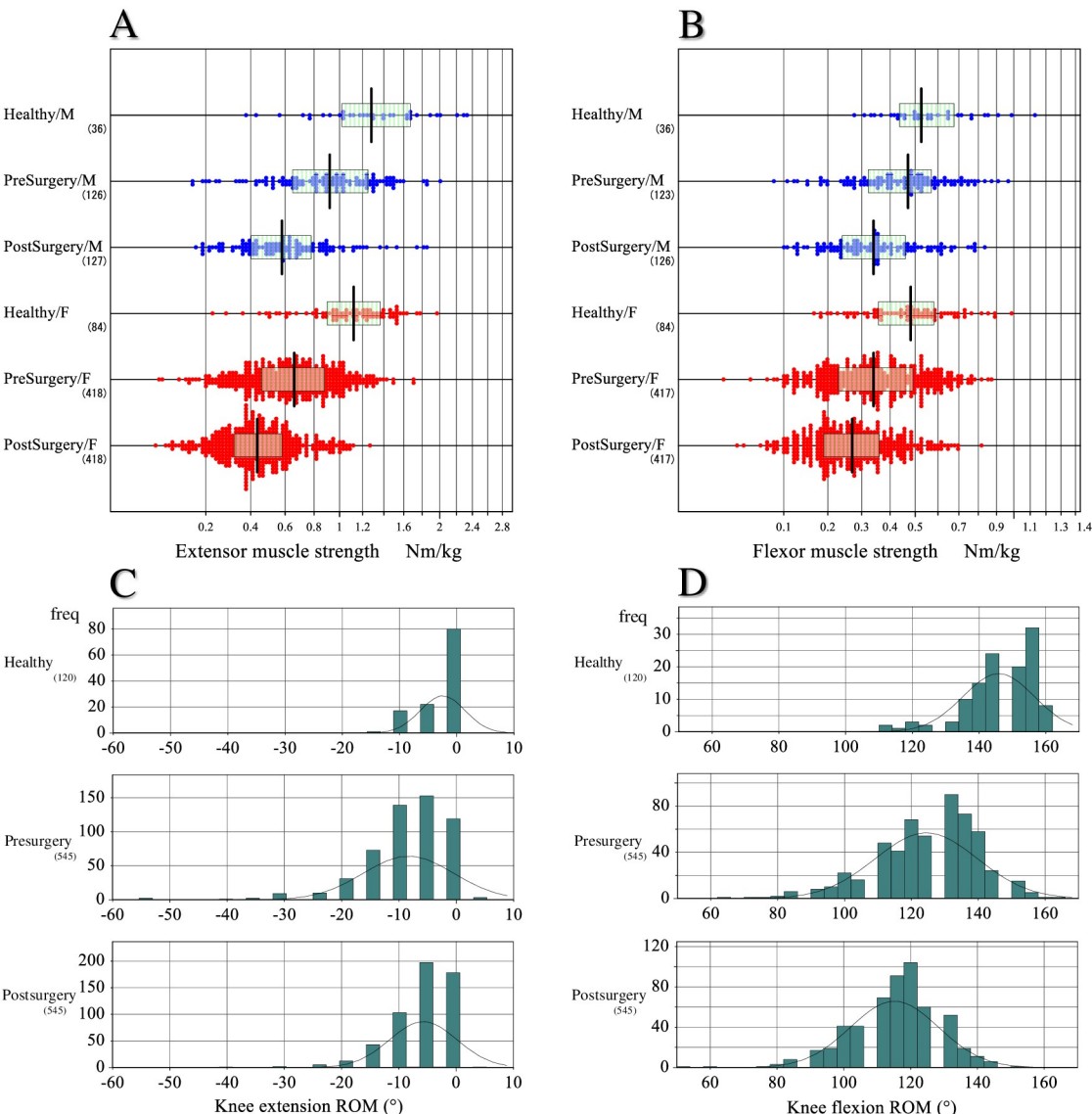

**Fig 4. Comparison of knee muscle strength and ROM between three groups.** The measured strengths of extensor (A) and flexor (B) muscles were compared between three groups: Healthy controls and patients before and after the surgery. The values were partitioned by gender. The x-axis was transformed to the power of 0.5 to make their distribution near Gaussian. The box and center line represent the mid-50% range and median of each subgroup, respectively. Similarly, the range of motion (ROM) of extension (C) and flexion (D) was compared between the three groups. As the test results were recorded discretely for every 5˚, the distribution is shown by histogram.

## Assessment of factors to derive RIs specific to subgroups

We considered sex, age, BMI, and surgical mode as possible factors for the need of subgrouping test results when deriving RIs. Three-level nested ANOVA was performed separately for pre- and post-surgical results by use of the following three factors of sex, age, and BMI for the former, and sex, age, and surgical mode for the latter. As shown in Table 3 by setting SDR ≥0.4 as a threshold, for values before the surgery, partitioning by age was required for TUG-com, TUGmax, and MWS with respective SDRage values of 0.445, 0.462, and 0.407. However, partitioning by sex was required for extensor and flexor muscle strength with respective

**Table 3. SDRs calculated to assess the need for partitioning values.**

| | Before surgery | | | After surgery | | |
|---|---|---|---|---|---|---|
| | **SDRsex** | **SDRage** | **SDR_BMI** | **SDRsex** | **SDRage** | **SDRsur** |
| TUG comfort | 0.000 | **0.445** | 0.134 | 0.000 | **0.416** | 0.282 |
| TUG maximum | 0.000 | **0.462** | 0.201 | 0.000 | **0.456** | 0.298 |
| Maximum walking speed | 0.168 | **0.407** | 0.138 | 0.112 | **0.363** | **0.442** |
| Extensor muscle strength | **0.573** | 0.099 | 0.144 | **0.484** | 0.000 | **0.303** |
| Flexor muscle strength | **0.418** | 0.000 | 0.171 | **0.379** | 0.187 | 0.034 |
| Extension ROM | 0.000 | 0.000 | 0.159 | 0.173 | 0.000 | 0.159 |
| Flexion ROM | 0.174 | 0.000 | 0.252 | 0.000 | 0.000 | 0.263 |

Three-level nested ANOVA was performed in two ways. One for motor function tests before the surgery was performed by setting sex, age, and BMI as factors of variation, and the other for motor function tests after the surgery was performed by setting sex, age, and surgical mode as the factors. In the analyses, age and BMI were respectively partitioned at 70 and 80 years and at 18.5, 25, and 30 kg/m$^2$. SDs attributable to sex, age, BMI, and surgical modes (SDsex, SDage, SDBMI, and SDsur) were divided by SD between-individuals (SDindiv) to obtain SDRsex, SDRage, SDRBMI, and SDRsur: i.e., SDindiv represents the residual SD after subtracting the influence of other sources of variation. We regard SDR $\geq$0.4 as a guide to judge the need for partitioning test results into subgroups when deriving the RIs [11].

SDRsex values of 0.573 and 0.418, implying a more prominent sex difference in extensor than flexor muscle strength.

In contrast, for values after the surgery, partitioning by age was required again for TUGcom and TUGmax. However, for MWS, SDR for surgical mode (SDRsur) was greater than SDRage. Therefore, we chose to partition the postsurgical MWS values into three categories by the surgical modes of UKA, MIS-TKA, and C-TKA. For post-surgical extensor and flexor muscle strength, the SDRsex values (0.484 and 0.379, respectively) were mixed relative to the threshold of 0.4. However, we chose to partition by sex both extensor and flexor muscle values for the sake of consistency with the pre-surgical values. As the SDRs for extension and flexion ROM were all below the threshold, no attempt to partition by sex and age was made for the values of these ROMs.

## Determination of RIs

RIs for motor function test results were derived according to the scheme shown above guided by SDR values. The RIs derived are listed in seven blocks in Table 4. The parametric method was basically used in the derivation. The success of the Gaussian transformation of test results was confirmed by Kolmogorov-Smirnov test as shown in Table 4 and in S1 Fig. The nonparametric method of computing the 2.5–97.5 percentile range was used in determining RIs for knee extension and flexion ROMs because their angle values were recorded discretely at every 5˚ and were highly skewed to the lower sides as shown in Fig 4C and 4D.

The RIs determined from healthy controls were not partitioned by sex or by age because of the small data size of 120 samples. The data size (n) shown in the table fluctuates because the parametric method performed truncation once at mean ± 2.81 SD under the transformed scale. The derivation process of the motor function tests before the surgery (marked as *1) are documented in S1 Fig to show the success of the Gaussian transformation by use of the probability plot and Kolmogorov-Smirnov test (N.S. implies no significant deviation from the Gaussian shape).

## Discussion

Physical therapists perform peri-surgical rehabilitation of KOA patients undergoing knee arthroplasty in collaboration with orthopedic surgeons. For optimal management and

**Table 4. Reference intervals derived for motor function parameters.**

**TUG comfort**

| Group | Sex | Age | n | Parametric RIs LL | Me | UL | K-S test |
|---|---|---|---|---|---|---|---|
| Healthy | All | All | 118 | 5.1 | 6.5 | 8.6 | 0.330 |
| PreSurg *1 | All | -69 | 118 | 7.6 | 11.2 | 20.6 | 0.635 |
| | All | 70–79 | 281 | 8.0 | 12.9 | 24.8 | 0.940 |
| | All | 80- | 140 | 8.9 | 15.5 | 33.6 | 0.730 |
| PostSurg | All | -69 | 119 | 8.1 | 12.3 | 27.4 | 0.150 |
| | All | 70–79 | 279 | 9.2 | 15.2 | 37.2 | 0.090 |
| | All | 80- | 140 | 10.3 | 18.7 | 38.6 | 0.970 |

**Extensor muscle strength**

| | Sex | n | Parametric RIs LL | Me | UL | K-S test |
|---|---|---|---|---|---|---|
| Healthy | All | 120 | 0.37 | 1.16 | 2.00 | 1.000 |
| PreSurg *1 | M | 124 | 0.25 | 0.91 | 1.88 | 1.000 |
| | F | 404 | 0.20 | 0.65 | 1.36 | 0.993 |
| PostSurg | M | 127 | 0.20 | 0.57 | 1.46 | 1.000 |
| | F | 417 | 0.14 | 0.40 | 0.82 | 0.530 |

**Flexor muscle strength**

| | Sex | n | Parametric RIs LL | Me | UL | K-S test |
|---|---|---|---|---|---|---|
| Healthy | All | 120 | 0.21 | 0.49 | 0.93 | 1.000 |
| PreSurg *1 | M | 121 | 0.14 | 0.45 | 0.88 | 1.000 |
| | F | 403 | 0.11 | 0.34 | 0.79 | 0.332 |
| PostSurg | M | 126 | 0.13 | 0.34 | 0.79 | 0.580 |
| | F | 416 | 0.09 | 0.26 | 0.61 | 0.900 |

**TUG maximum**

| Group | Sex | Age | n | Parametric RIs LL | Me | UL | K-S test |
|---|---|---|---|---|---|---|---|
| Healthy | All | All | 119 | 3.9 | 5.0 | 7.2 | 0.160 |
| PreSurg *1 | All | -69 | 118 | 5.9 | 8.7 | 17.0 | 0.585 |
| | All | 70–79 | 279 | 6.4 | 10.3 | 18.3 | 0.191 |
| | All | 80- | 138 | 6.7 | 12.6 | 23.4 | 0.810 |
| PostSurg | All | -69 | 118 | 6.4 | 9.9 | 20.1 | 0.070 |
| | All | 70–79 | 279 | 7.7 | 12.2 | 29.1 | 0.060 |
| | All | 80- | 139 | 8.0 | 15.0 | 28.1 | 1.000 |

**Extension ROM**

| | Sex | n | Nonparametric RIs LL | Me | UL |
|---|---|---|---|---|---|
| Healthy | All | 120 | 0 | 0 | −10 |
| PreSurg | All | 545 | 0 | −5 | −30 |
| PostSurg | All | 545 | 0 | −5 | −20 |

**Maximum Walking Speed (MWS)**

| Group | | Sex | Age | n | Parametric RIs LL | Me | UL | K-S test |
|---|---|---|---|---|---|---|---|---|
| Healthy | | All | All | 120 | 1.50 | 2.20 | 3.13 | 0.630 |
| PreSurg *1 | | All | -69 | 120 | 0.60 | 1.22 | 2.03 | 0.420 |
| | | All | 70–79 | 285 | 0.45 | 1.07 | 1.73 | 0.850 |
| | | All | 80- | 140 | 0.43 | 0.90 | 1.59 | 0.856 |
| PostSurg | UKA | All | All | 104 | 0.55 | 1.00 | 2.01 | 1.000 |
| | MIS-TKA | All | All | 342 | 0.37 | 0.89 | 1.61 | 0.740 |
| | C-TKA | All | All | 99 | 0.27 | 0.74 | 1.47 | 0.230 |

**Flexion ROM**

| | Sex | n | Nonparametric RIs LL | Me | UL |
|---|---|---|---|---|---|
| Healthy | All | 120 | 120 | 150 | 160 |
| PreSurg | All | 545 | 90 | 125 | 150 |
| PostSurg | All | 545 | 85 | 115 | 140 |

RIs were derived for motor function parameters for healthy controls, patients before and after surgery (PreSurg and PostSurg). Partitioning by age was done for TUGs and MWS, and by sex for extensor and flexor muscles strength. No partitioning by sex and age was done for values of healthy controls due to the small data size. RIs were determined by nonparametric method for test results of ROMs. For all other motor function tests, RIs were derive by parametric method through Gaussian transformation of test results (see main texts). The appropriateness of the transformation was checked by use of Kolmogorov-Smirnov (K-S) test: i.e., P>0.05 in the rightmost column was regarded as successful transformation. Refer to S1 Fig for the parameters marked as (*1) for actual outputs for the transformation and K-S test.

promotion of recovery, it is important to monitor knee motor functions such as TUG, MWS, knee muscle strength, and knee ROM. For objective use of these parameters, it is essential to have reliable RIs for them and to analyze their sources of variation.

Regarding RIs, there are several studies reporting 'normative' values for knee motor functions derived from healthy individuals. Tsubaki et al. [28] measured TUG in 172 healthy Japanese elderly (80 men, 92 women; 50–79 years of age) and showed age-specific reference values. Yoshimura et al. [29] reported reference values for MWS from the analyses of 2468 healthy Japanese elderly (826 men, 1642 women; mean age 71.8 years). Seino et al. [9] also reported healthy values for MWS. However, all the past studied reported values in mean ± SD format

without consideration of the distribution pattern. Therefore, it is not possible to predict the RI or central 95% range of value from the reported values.

In addition to these methodological issues of the past studies targeting healthy subjects, as demonstrated in Figs 3 and 4, healthy ranges of knee motor function tests were quite different from those of KOA patients both before and after the surgery. Therefore, our team of physical therapists conducted this study with a main objective of exploring the sources of variation and determining RIs for knee motor function tests before and after the surgery by use of up-to-date statistical methods currently in use in the field of laboratory medicine [10, 11].

Analysis of the sources of variation of the motor function tests before surgery clearly showed that both TUG comfort and TUG maximum in our KOA patients increased with age. Steffen et al. reported that TUG maximum of healthy elderly subjects increased proportionately with age [30]. Adegoke et al. [31] studied TUG maximum and MWS in KOA patients and reported a similar age dependency and attributed it to unbalanced walking with age. Bohannon [15] reported in his study of healthy elderly subjects that compared to comfortable walking speed, the maximum walking speed decreased prominently with age. Knee muscle strength before the surgery in our patients was lower in the women. Logerstedt et al. [32] reported the same finding in their KOA patients before surgery.

In contrast, in the motor function tests performed 14 days after the surgery, the post-surgical increase in both TUG comfort and TUG maximum was more pronounced with age, but the increase was suppressed in patients who received UKA. Jones et al. [33] reported that UKA led to reduced TUG maximum because of limited surgical involvement of the quadricep muscles. Similarly, post-surgical MWS was decreased with age and influenced by the surgical mode. Meanwhile, extensor and flexor muscle strengths after surgery were lower in the women, as was already reported by Gustavson et al. [34] in KOA patients. Meanwhile, we found that the extensor muscle strength was also influenced by surgical mode with UKA resulting in higher strength than C-TKA.

We relied on the SDR to judge the need for partitioning test results in determining RIs specific for certain subgroups. It was not difficult to judge SDR for pre-surgical motor functions because either sex or age was the only factor to be considered. However, for post-surgical parameters, we had three factors to consider: sex, age, and surgical mode. For example, TUG comfort and TUG maximum after surgery were both associated with age and surgical mode. However, because of the insufficient sample size, we had to adopt just one factor with a higher SDR to avoid too many partitions. Therefore, we partitioned post-surgical TUG values only by age into three groups to compute the RIs.

Post-surgical MWS also showed significant association with both age and surgical mode, but we made the opposite decision of partitioning MWS values by surgical mode rather than by age based on the values of SDR for age = 0.363 versus SDR for surgical mode = 0.442. As the rationale for this decision, we found a report by Iijima et al. [35] showing that in KOA patients, turn-around speed during the TUG test depended on hip abductor muscles that are minimally affected by the surgery, whereas straight-walking speed depended mostly on the strength of the quadriceps. Therefore, selection among the three surgical modes with different degrees of quadriceps injury had more influence on MWS that represents straight-walking speed.

For partitioning of the RIs for extensor and flexor muscle strength, sex with high SDR was the only factor requiring consideration. This finding of sex-difference was supported in a report by Logerstedt et al. [32]. Regarding the source of variation of the ROM, we found the levels of flexion ROM were negatively associated with BMI before surgery, and their post-surgical values were better with UKA. However, the actual differences in terms of SDR were less than the threshold of 0.4, and thus, their RIs were determined without any partitioning.

In this study we showed that both pre- and post-surgical knee motor functions were all conspicuously different from those of the healthy elderly subjects as shown in Fig 3 for TUG and MWS and in Fig 4 for muscle strength and knee ROM. The obvious gaps we observed in all parameters from healthy elderly subjects clearly point to the importance of disease specific RIs for knee arthroplasty patients.

In summary, the clinical implication of this study is as follows. As we only enrolled patients who followed a normal course of recovery, the RIs stratified according to their factors of variation will facilitate peri-surgical rehabilitation customized to patients according to sex, age, and surgical mode. Furthermore, the subgroup-specific RIs will be useful in detecting patients who deviate from the normal course of recovery.

## Limitations

This study has three limitations. 1) The sample size was not large enough to make a finer subgrouping of test results by sex, age, and surgical mode. We were obliged to partition tests results only by a single factor from these three major factors. 2) For the long-term management of knee arthroplasty patients, knee motor function tests should be monitored for a longer period. However, this was beyond the scope of our study. Even in routine clinical practice, the long-term follow-up of patients' motor functions is difficult on the physical therapy side because most patients return to local clinics after their surgery. 3) We only evaluated the knee motor function tests that can be safely performed in the early post-surgical period. For long-term evaluation of knee arthroplasty patients, the 6-minute walk test and stair climb test are available. However, these tests were not included in our evaluation because of difficulty in performing them during the immediate post-surgical period.

## Conclusion

With the increasing demand of knee arthroplasty for KOA patients, there is a dire need for objective reference values for knee motor functions to ensure efficient peri-surgical management of patients. Through the collaboration of 13 medical institutions in Japan that specialize in knee arthroplasty, we enrolled 545 KOA patients undergoing knee arthroplasty, who followed a normal course of recovery, and performed knee motor functions tests twice before and 14 days after the surgery. RIs specific to each time point were determined in consideration of the distribution pattern of each parameter. The need for partitioning of RIs was objectively judged by use of an SDR that represents the effect size of between-subgroup differences. To our knowledge, this is the first report in the field of orthopedics and rehabilitation to determine RIs specific to KOA patients by use of up-to-date statistical methods. The quantitative information on RIs and their factors of variation will facilitate objective implementation of peri-surgical rehabilitation and screening of patients who deviate from the normal course of recovery.

## Supporting information

**S1 Fig. Determination of reference intervals by parametric method.**
(PDF)

**S1 File.**
(XLSX)

## Author Contributions

**Conceptualization:** Hideyuki Ito, Kotaro Tamari, Tetsuya Amano.

**Data curation:** Hideyuki Ito, Kotaro Tamari, Tetsuya Amano, Shigeharu Tanaka, Shigehiro Uchida, Shinya Morikawa.

**Formal analysis:** Hideyuki Ito, Kiyoshi Ichihara.

**Investigation:** Hideyuki Ito.

**Methodology:** Hideyuki Ito, Kiyoshi Ichihara, Tetsuya Amano, Shigeharu Tanaka, Shigehiro Uchida.

**Project administration:** Hideyuki Ito.

**Resources:** Hideyuki Ito.

**Supervision:** Kiyoshi Ichihara, Kotaro Tamari, Tetsuya Amano, Shigeharu Tanaka.

**Validation:** Hideyuki Ito, Kiyoshi Ichihara, Kotaro Tamari, Shigehiro Uchida, Shinya Morikawa.

**Writing – original draft:** Hideyuki Ito, Kiyoshi Ichihara.

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
