## [Decision Letter · Decision Letter 0]

18 Dec 2020

PONE-D-20-18248

Determination of reference intervals for lower-limb motor functions specific to patients undergoing knee arthroplasty

PLOS ONE

Dear Dr. Ichihara,

Thank you for submitting your manuscript to PLOS ONE. After careful consideration, we feel that it has merit but does not fully meet PLOS ONE’s publication criteria as it currently stands. Therefore, we invite you to submit a revised version of the manuscript that addresses the points raised during the review process.

Please address the insightful points identified by the reviewers below.

We look forward to receiving your revised manuscript.

Kind regards,

Alison Rushton

Academic Editor

PLOS ONE

Journal Requirements:

Reviewers' comments:

Reviewer's Responses to Questions

**Comments to the Author**

1. Is the manuscript technically sound, and do the data support the conclusions?

Reviewer #1: Partly

Reviewer #2: Yes

Reviewer #3: Partly

2. Has the statistical analysis been performed appropriately and rigorously? 

Reviewer #1: Yes

Reviewer #2: Yes

Reviewer #3: Yes

3. Have the authors made all data underlying the findings in their manuscript fully available?

Reviewer #1: No

Reviewer #2: Yes

Reviewer #3: No

4. Is the manuscript presented in an intelligible fashion and written in standard English?

Reviewer #1: Yes

Reviewer #2: No

Reviewer #3: Yes

5. Review Comments to the Author

Reviewer #1: 

The manuscript is generally well structured and written in precise and understandable manner, but some points still need clarifications:

1. Please include information on timing of patients' examinations in the method section. It is for the first time given in the Discussion.

2. Line 181: “objective variable” is an unusual name. “Dependent variable” is more typical.

3. What is the meaning of the phrase “large data size” in line 188? Do you mean the number of variables or the number of cases? If the number of cases, then it somewhat contradicts limitations of the study. Similarly, what is meant by “small data size” in caption to Table 3?

4. “Probability paper plot” is typically called just “probability plot” or “Q-Q plot” (line 227 and others).

5. Have the KOA patients’ and healthy controls’ baseline characteristics been compared by statical tests? If so, such comparisons should be shown.

6. Why p-values are not used as a measure of significance of effects either in multiple regression or ANOVA? The should be reported in supplementary material at least to show if observed effects were significant.

7. Please reconsider graphical presentation in Fig. 1. Although presentation of raw data point is a good practice, in some cases they make your box plots difficult to read. I would recommend presentation that emphasizes the boxes more and points less (the one in Fig. 2 is better). Besides, panels are referenced by letters that are absent in the figure (also in Fig. 2).

8. Line 306 states that “Although the data are not shown, the level of decrease in muscle strength did not depend on the surgical mode but seemed to be simply due to immobility after surgery”. Ideally such data should be shown, at least in supplementary file.

9. Fig. 2, lower panels: vertical axis scale and legend is missing.

10. Caption to table 2: symbol for between-individual variation is different that one used in the methodology section. “SDsex” should probably be replaced by “SDRsex”.

11. Suppl. Fig. 2: test results (but not necessarily figures) showing successful transformation of distributions should be shown for all variables, ideally after partitioning as this is crucial factor for the choice of reference interval calculation method. If groups resulting from partitioning are not numerous enough for Kolmogorov-Smirnov test, then Shapiro-Wilk test can be used.

12. Fig. 3: what is the value of parameter p in left panel? The value of 0, that is given in the figure, would result in division by 0.

13: Line 363: description of non-parametric method of reference interval derivation should be moved to methods.

14. Table 3: tilde (~) symbol is quite confusing in the table. Usage of lower than, greater than and hyphen signs would be clearer.

Reviewer #2: 

The study presents the results of an original research, however some corrections must be taken into account.

- In the abstract, make the results clearer and in the conclusion specify how the information can contribute to the management of surgical management of KOA patients.

- It is necessary that the introduction is more robust, so that the clinical importance of the study is clear.

- There is no need to have a discussion of statistics in the introduction.

-Leave the objective of the study, highlighting the clinical relevance of the study

-Rows 64-66: I suggest deleting the information or placing a reference.

-Line 85: Delete the phrase "we found ir necessary ..." all information should be based only on the literature.

- In the first paragraph of the discussion, present the main results of this study.

-The authors in every manuscript, describe a lot of information and discussions about statist tests, however it is important to facilitate the reading so that clinicians can use these information to carry out evidence-based practice. I suggest making changes in the discussion, clarifying the importance of these findings, and clinical significance.

-Review English

Reviewer #3: 

1. Is the manuscript technically sound, and do the data support the conclusions?

The rationale could be clearer, and while overall the statistical techniques are sound I am unconvinced the data have been adequately discussed. I have some specific criticisms outlined below.

2. Has the statistical analysis been performed appropriately and rigorously?

Overall, the statistical techniques are sound. But I have some specific criticisms outlined below

*3. Have the authors made all data underlying the findings in their manuscript fully available?

No, not all data is available but is upon request to the corresponding author.

*4. Is the manuscript presented in an intelligible fashion and written in standard English?

There are suggested edits which are outlined in the comments to authors below. I commend the authors in producing the manuscript if English is not their first language. Overall the manuscript would benefit from some simplifying and accuracy in language and layout including a final sense-check from someone who writes in English as a first language. I provide some typos and edits below.

*5. Review Comments to the Author

The authors provide a technical investigation of normal recovery profiles in surgical interventions for KOA with respect to function (TUG) and impairment (strength and ROM) measurements that are feasible to measure in the clinic, and the associated effect of factors (age, gender etc) pre operatively and 14-days post operatively. Overall, I found the manuscript difficult to read and I think this is because the rationale was not entirely clear, and there is no clear mapping of objectives to the results and discussion sections. Based on this, I think the manuscript requires re-writing. I offer more detailed feedback below.

Please see attached document for more detailed comments for the authors.

6. PLOS authors have the option to publish the peer review history of their article (what does this mean?). If published, this will include your full peer review and any attached files.

Reviewer #1: No

Reviewer #2: No

Reviewer #3: **Yes: **Gareth D. Jones

---

## [Author Response · Author response to Decision Letter 0]

2 Jan 2021

Reviewer #1

Our response ⇒ We appreciate a great deal for your time and effort to review our manuscript and for providing us with invaluable comments to improve our manuscript. We carefully addressed each issue and revised the manuscript accordingly as describe one by one below.

1. Please include information on timing of patients' examinations in the method section. It is for the first time given in the Discussion.　Line426

Our response ⇒ We described the timing for the post-surgical measurements of the knee motor functions in the Methods as follows: “The measurements of the following knee motor functions were performed twice in each KOA patients just before and at two weeks after knee arthroplasty.”.

2. Line 181: “objective variable” is an unusual name. “Dependent variable” is more typical.

Our response ⇒ We changed “objective variable” to “dependent variable” as pointed out.

3. What is the meaning of the phrase “large data size” in line 188? Do you mean the number of variables or the number of cases? If the number of cases, then it somewhat contradicts limitations of the study. Similarly, what is meant by “small data size” in caption to Table 3?

Our response ⇒ We meant that testing significance of correlations (like rp) with a large sample size (like n>200), even small rp can be statistically significant. However, we chose to show P values for each rp (as may be expected by the readers). Therefore, we changed the original description “Because of the large data size, the “effect size” of the 189 standardized partial regression coefficient (rp) was set to |rp|≥0.2” to “In addition to expressing the statistical significance of each standardized partial regression coefficient (rp) in P value, we marked its effect size (practical significance) by use of a threshold of |rp|≥0.2”

4. “Probability paper plot” is typically called just “probability plot” or “Q-Q plot” (line 227 and others).

Our response ⇒ We changed the term as kindly advised. 

5. Have the KOA patients’ and healthy controls’ baseline characteristics been compared by statical tests? If so, such comparisons should be shown.

Our response ⇒ Following the advice, we newly made Table 1 that listed the demographic information comparing KOA patients and healthy subjects. The statistical testing results were also included in the table. 

6. Why p-values are not used as a measure of significance of effects either in multiple regression or ANOVA? The should be reported in supplementary material at least to show if observed effects were significant.

Our response ⇒ We followed the advice and revised the table of multiple regression analysis (MRA) by adding judgment based on P-value for each rp (standardized partial regression coefficient, which corresponds “effect size” the correlatoin). 

7. Please reconsider graphical presentation in Fig. 1. Although presentation of raw data point is a good practice, in some cases they make your box plots difficult to read. I would recommend presentation that emphasizes the boxes more and points less (the one in Fig. 2 is better). Besides, panels are referenced by letters that are absent in the figure (also in Fig. 2).

Our response ⇒ We redrew the 1D scattergrams with the central 50% range (box) and vertical median line drawn last so that they won’t be hidden behind the dots. Please note that we added another figure as Fig. 1 and thus original Fig.1 and 2 are now Fig. 2 and 3.

8. Line 306 states that “Although the data are not shown, the level of decrease in muscle strength did not depend on the surgical mode but seemed to be simply due to immobility after surgery”. Ideally such data should be shown, at least in supplementary file.

Our response ⇒ We realized that the above expression was not appropriate as was also pointed out by Reviewr-3, we simply deleted the confusing/speculative remark. 

9. Fig. 2, lower panels: vertical axis scale and legend is missing.

Our response ⇒ We thankfully amended Fig. 2 (now Fig. 3) accordingly.

10. Caption to table 2: symbol for between-individual variation is different that one used in the methodology section. “SDsex” should probably be replaced by “SDRsex”.

Our response ⇒ We corrected “SDbi” to “SDindiv”, and “SDsex” to “SDRsex” as pointed.

11. Suppl. Fig. 2: test results (but not necessarily figures) showing successful transformation of distributions should be shown for all variables, ideally after partitioning as this is crucial factor for the choice of reference interval calculation method. If groups resulting from partitioning are not numerous enough for Kolmogorov-Smirnov test, then Shapiro-Wilk test can be used.

Our response ⇒ We followed the suggestion and we added one rightmost column to Table 4 (old Table 3, listing derived RIs) to show Kolmogorov-Smirnov (K-S) test results as P-value. 

12. Fig. 3: what is the value of parameter p in left panel? The value of 0, that is given in the figure, would result in division by 0.

Our response ⇒ We are sorry about the confusing Fig. 3. We chose to delete it because of the following reason: The value of ‘p’ (lower-case p) represented “power” used in the Box-Cox power transformation. If p=0.0, it indicates logarithmic transformation, and if p=0.5, square-root transformation, etc. However, the figure was made as an illustrative example and the value of p (0.0, 0.5, and 0.3) was set approximately and true values of p was in Suppl Fig. 1. Therefore, we found it confusing and not worth showing in this manuscript. 

13: Line 363: description of non-parametric method of reference interval derivation should be moved to methods.

Our response ⇒ We are sorry for the omission of defining nonparametric method. We added following explanation in the Methods: “When test results of a given parameter failed in the Gaussian transformation by the Box-Cox method or when test results took discrete values, we applied nonparametric method for determining the central 95% range of test results after sorting them, and took the range of 2.5 and 97.5 percentiles as the limits of the RI.”

14. Table 3: tilde (~) symbol is quite confusing in the table. Usage of lower than, greater than and hyphen signs would be clearer.

Our response ⇒ As suggested, we replaced (~) to (-) 

Reviewer #2

- In the abstract, make the results clearer and in the conclusion specify how the information can contribute to the management of surgical management of KOA patients.

- It is necessary that the introduction is more robust, so that the clinical importance of the study is clear.

Our response ⇒ We are grateful for your time and kind effort to review our manuscript and to give us invaluable comments. For the first advice, we were pointed out the same problems and thus extensively amended our manuscript especially for the Introduction and Discussion. We made our best to clarify the background and objective of this study and make clear of the clinical implication of our results. 

- There is no need to have a discussion of statistics in the introduction.

Our response ⇒ As mentioned above, in the process of rewriting the Introduction, we deleted the statistical issues in the past normative value studies and move them to the Discussion. 

-Leave the objective of the study, highlighting the clinical relevance of the study

Our response ⇒ We believe after rewriting, clinical relevance of this study from the perspective of rehabilitation medicine has become clearer by revising the descriptions at the end of both the Introduction and the Discussion.

-Rows 64-66: I suggest deleting the information or placing a reference.

Our response ⇒ We deleted that part of the Introduction as suggested.

-Line 85: Delete the phrase "we found it necessary ..." all information should be based only on the literature.

Our response ⇒ We deleted that description in the process of rewriting the Introduction. 

- In the first paragraph of the discussion, present the main results of this study.

Our response ⇒ We are sorry about the inappropriate description at the beginning of the Discussion. We rewrote the first paragraph by recalling the objectives of this study and then briefly wrote about the main results. 

-The authors in every manuscript, describe a lot of information and discussions about statist tests, however it is important to facilitate the reading so that clinicians can use these information to carry out evidence-based practice. I suggest making changes in the discussion, clarifying the importance of these findings, and clinical significance.

Our response ⇒ Again, we apologize for poor writing of the manuscript without giving clear focus on the clinical implication of this study. After rewriting, we made it clear that this study originated from physical therapists who are in charge of rehabilitation of KOA patients undergoing knee arthroplasty. We then emphasized the importance of reference intervals of knee motor function tests for improved/objective implementation of peri-surgical rehabilitation, that are supposed to contribute to and predict the long-term success of the surgery. 

Reviewer #3

Reference intervals of knee motor function tests for patients undergoing knee arthroplasty 

The authors provide a technical investigation of normal recovery profiles in surgical interventions for KOA with respect to function (TUG) and impairment (strength and ROM) measurements that are feasible to measure in the clinic, and the associated effect of factors (age, gender etc) pre operatively and 14-days post operatively. Overall, I found the manuscript difficult to read and I think this is because the rationale was not entirely clear, and there is no clear mapping of objectives to the results and discussion sections. Based on this, I think the manuscript requires re-writing. I offer more detailed feedback below.

Please see attached document for more detailed comments for the authors.

Further Review Comments to the Author The authors provide a technical investigation of normal recovery profiles in surgical interventions for KOA with respect to function (TUG) and impairment (strength and ROM) measurements that are feasible to measure in the clinic, and the associated effect of factors (age, gender etc) pre operatively and 14-days post operatively. Overall, I found the manuscript difficult to read and I think this is because the rationale was not entirely clear, and there is no clear mapping of objectives to the results and discussion sections. Based on this, I think the manuscript requires re-writing. I offer more detailed feedback below.

Our response ⇒ We are indeed grateful for your time and effort to critically review our manuscript and for providing us with invaluable comments to improve our manuscript. We carefully addressed each issue and revised the manuscript accordingly as describe one by one below.

Introduction & Rationale

I sense the focus of the paper is in its utilisation of deploying statistical techniques to derive RIs that addressed the problem of non-normal data distributions. What I think is lacking in the introduction and discussion is a clear clinical rationale that this will help clinical practice. I agree that clinicians would welcome predicted recovery trajectories of typical KOA patients undergoing surgery based on the variables, but fear that these data alone might not affect practice due to the many other factors under consideration that the authors do not acknowledge e.g. fear of movement post-surgery, smoking and lifestyle choices. I could not see a clear statement in the introduction nor discussion that defends why this (albeit large) sample is definitively representative of the patient population in Japan. There is no information detailing the recruitment process for either group or justification for the discrepancy between the two groups sample size. The data do offer a basis on which studies could now be developed to explore why patients might deviate from a “normative” presurgical state and a “normative” recovery, and for that this paper is welcome. But I think the paper could benefit from a specific allusion to this and to how this relates or interacts with “healthy” age-matched adults.

Our response ⇒ We admit the problem of unclear description on our objectives of this study. We rewrote the section at the bottom of the Introduction to describe the background and objective more clearly, with suggestion of practical/clinical implications of this study.

To provide a scope of our subsequent responses, let us briefly list up the backgrounds as follow:

1) This is a research conducted by a team of physical therapists who are routinely involved in peri-surgical rehabilitation of KOA patients undergoing knee arthroplasty (KA) in close collaboration with orthopedic surgeons. [Please note that this study was not originated from orthopedicians].

2) For proper provision of rehabilitation to the patients, it was essential to have objective evaluations of knee-related motor functions before and after KA. 

3) However, the normative values (Means and SDs) for the motor functions reported in the past have been determined from test results obtained in healthy individuals by use of out-dated methods. Therefore, it was necessary to determine reference intervals (RIs) using up-to-date statistical methods from test results measured in KOA patients undergoing KA.

4) There are studied that reported the importance of knee motor functions short-term after KA in predicting the long-term success of KA, but no referrable RIs for that specific period exit in the literature. 

With these backgrounds, a team of physical therapists conducted this study with the following objectives: 

1) For objective use of knee motor function tests in the rehabilitation, to explore their sources of variation (SVs) and to determine their reference interval (RIs), which are specific to KOA patients undergoing KA. 

2) To apply up-to-date statistical methods in analyzing the SVs and in determining RIs, which have not been used in the field of rehabilitation medicine in the past,

(3) To compare knee motor functions before and two-weeks after knee arthroplasty, which is of relevance in predicting a long-term success of the surgery.

From these perspectives, we responded to the subsequent comments one by one and revised the section of Introduction extensively. 

With respect to the inquiry regarding the representativeness of 583 KOA patients, we recruited them consecutively from 13 major institutions in western Japan by use of a harmonized protocol, and thus we can regard them as reflective of current clinical practice of KA in Japn. Please refer to our response to the “Method” section below in Page 5 of this document. 

The introduction states that conservative therapy of KOA is the treatment of choice – I would like to read what conservative therapy is in this context, and what exactly defines epicondylar deformity to inform local surgical decisions so international readers can make a judgement on their own practice. A more explicit description of the clinical indicators for TKA surgery would strengthen the manuscript. 

Our response ⇒ We gave explanatory description on the nature of “conservative therapy of KOA” as “Conservative therapy, such as exercise, ultrasonic therapy, electrotherapy, orthosis therapy, is the primary choice”. For the TKA surgeries, we added explicit description of indications as follows: “with development of multiple osteophytes, sclerosis, or narrowing of joint space, knee arthroplasty becomes necessary” 

The authors cite 2 previous pieces of work (Bade et al 2014 & Mizner et al 2005) to defend the use of knee ROM, knee flexion/extension isometric strength and TUG by stating “The prognosis after these surgeries depends on the pre-operative and short-term post-operative capability of physical and motor functions” (L62-63). These two references do not in my opinion constitute an adequate defence of this statement by themselves and I would expect a much more compelling argument to be put forward regarding these particular measures seeing as they are the main basis for the paper. 

Our response ⇒ We realized that we needed to include more papers to argue the importance of short-term postsurgical motor functions. However, in the process of rewriting the Introduction, we chose not to give detailed description on those papers. Rather we wrote briefly as follows and placed the descriptions near the end of the Discussions: 

“Although we could only determine the RIs of knee motor function tests just before and at two weeks after the surgery, those peri-surgical values are of relevance in predicting the long-term outcome of KOA patients as reported by Dobson et al. [4]. The importance of pre-surgical knee motor functions in predicting the long-term recovery was also reported by Bade et al. [12] and Mizner et al. [13]. Furthermore, Bade et al. [12] and Zeni et al. [14] showed that values of TUG, knee muscle strength and knee flexion/extension ROM at short-term after surgery were important in predicting later recovery in daily activity. Therefore, we believe that the RIs derived in this study are of direct relevance in identifying patients whose motor functions deviate from those of others during the peri-surgical rehabilitation. Prompt attention and corrective measures would lead to improved speed of recovery from the surgery as suggested by Bade et al. [39] and Stevens et al. [40].”

References are missing for the section (L54-62) “Among the 25.3 million patients with KOA, 8 million have symptomatic disease, which poses a large socio-economic problem in Japan. KOA features degeneration and attrition of joint structure that result in ossification of the cartilage and surrounding tissue, which restricts knee movements and eventually leads to pain and gait disturbance. Conservative therapy is the primary choice, but with the development of deformity in one or both epicondyles, either unicompartmental knee arthroplasty (UKA) or total knee arthroplasty (TKA) has been the therapeutic regimen of choice. Recently, a less invasive regimen called minimally invasive surgery TKA (MIS-TKA) has become popular as an alternative to conventional TKA (C-TKA)”.

Our response ⇒ We are sorry about the inappropriate citation of the paper on epidemiology of KOA in Japan at the first paragraph of the Introduction. The descriptions were all based on a single paper [1]. We rephrased the parts as follow: “The prevalence of knee osteoarthritis (KOA) is as high as 42.6% in men and 62.4% in women above 40 years of age as reported in a large cohort study conducted in Japan in 2011 by Yoshimura et al. [1]. The report also disclosed that among the 25.3 million patients with KOA, 8 million have symptomatic disease, which poses a large socio-economic problem in Japan”.

The authors allow an entire paragraph (L67-78) listing previous TUG normative work based on healthy adults citing mean and SD values. Further explanation as to why this is specifically relevant to this study would be beneficial. I note though that the normative TUG meta-analysis work by Bohannan is missing from these data, and indeed from the whole paper (Bohannon, 2006). This is strange as not only did this work take various human ethnicities into account, it also included data from disparate TUG instructions – mainly to perform TUG at “comfortable” or “self-selected” pace or “quickly”. Despite these disparities in method the observed summary data was homogenous in Bohannon’s work. 

Our response ⇒ We thanks for pointing us out of the important paper. However, in the process of rewriting the Introduction to clarify our background and objectives in a concise manner, we deleted most of the past papers on normative values of knee functions. We rather moved them to the Discussion and thankfully cited the key paper of meta-analysis by Bohannon.

L79 – if a sample distribution is Gaussian I agree that a sample 95%RI could be expressed as mean ± 1.96SD, but would a population estimate be more useful (i.e. mean ± 1.96SE) yielding a mean and 95%CI? 

Our response ⇒ We understand that, if we are interested in comparing means between subgroups, the estimation of mean ±1.96SE is useful for the purpose, because SE represents standard error of mean: i.e., by-chance variability of mean. However, to determine the RI, it is necessary to know the variability of individual points (SD), not the variability of means (SE). If the distribution of values is Gaussian, we could calculate the RI as mean ± 1.96SD. If not, we first need to convert values to make the distribution Gaussian before calculating the RI.

As an aside comment - I was interested to note that TUG undertaken at a comfortable and maximal tempo was included in the current study. The authors did not as far as I could see justify why this would be helpful and did not explain if the two variants in instructions were undertaken randomly. I think dealing with these two criticisms would benefit the manuscript. I am guessing the debate about maximum versus comfortable comes from the disparity in instruction language. According to some literature, for example (Kamide et al., 2011), the original TUG paper (Podsiadlo and Richardson, 1991) instructed participants to perform the TUG at a usual or comfortable pace, whereas a later protocol (Shumway-Cook et al., 2000) asked participants to perform TUG with maximum effort. This is an understandable but, in my opinion, incorrect interpretation of there being disparate protocols and I think alluding to this disparity is unhelpful. 

This is because first, the Shumway-Cook paper asked participants to perform TUG “quickly and safely”, not “with maximal effort” nor “as fast as possible”. Second, I think Shumway-Cook and colleagues included a temporal qualifier (“quickly”) simply because participants were instructed to perform TUG under multiple dual-task conditions in their experiment. It is of course possible that within-subject TUG performance can be influenced by instructions to perform the task either comfortably or maximally, so there is a practical reason to include both protocol phenotypes in determining reference values as did the authors in the paper under review but only if there is a debate in the community about how to instruct the TUG. My interpretation is there is no such debate among clinicians. Even so, if a strong case could be constructed in this manuscript, then I would expect the two protocols to have been undertaken by an independent group or sufficiently apart and randomly by the same individuals to avoid any interaction effects. In contrast, it is interesting that forward average walking velocity was only measured at maximal tempo in the authors’ paper. Why not comfortable pace as well?? I would also like to see the authors defend the use of maximal walking velocity within the context of average self-selected walking velocity parameters (step width and length) being observed to be less variable than velocities under maximal volition (Sekiya et al., 1997).

Our response ⇒ We understood the importance of distinguishing “comfortable TUG” from “maximum TUG”, which depends on the nature of instructions to be given to the examinees, which tend to differ depending on the researchers. Accordingly, we clarified in the Methods which type of TUG test was used in each paper we cited. In our study, we performed TUG both at comfortable speed as originally proposed by Podsiadlo and at the maximum speed as recommended by Shumway-Cook. On the other hand, for measuring walking speed, we just adopted the maximum speed because Dbson et al. recommended it due to higher precision of measurement. We cited those references in describing TUG and MWS.

Overall, I think the introduction lacks some clarity and would benefit from a more compelling argument for why this work is needed, and what it aimed to add for clinical practitioners or researchers.

Our response ⇒ As we described above, we rewrote the Introduction according to the helpful comments.

Method

Patients – the sample size is large (n>500), which is an advantage, across 13 institutions which is a disadvantage without some description of differences in patients and clinical practice between them so readers can make an informed decision about the validity of the study. For example, while the inclusion criterion of indication for surgery is sensible it would be good to know if the different institutions deployed similar clinical reasoning in their decision making and whether patients were included on that decision or not. 

Our response ⇒ As we responded above, we regard the 583 KOA patients [Note: the number was originally 545 which was after applying exclusion criteria] undergoing knee arthroplasty (KA) are representative of the current practice for KA in Japan because we recruited them consecutively from 13 major institutions located in a wide area of western Japan. The protocol of the study was elaborated in harmonized ways. We modified our description on the patients as follows: 

“A total of 583 KOA patients undergoing elective knee arthroplasty were recruited consecutively between July 2013 and February 2018 by use of harmonized study protocol from 13 institutions specialized in knee arthroplasty, which are scattered widely in western Japan.”

The allusion from L107 that a standard clinical rehabilitation is shared across the institutions is welcome, but could be moved to be more prominent within the paragraph. Additionally, I would like to see a better breakdown of what the specification of the rehabilitation programme is here in addition to L109-112, either tabulated or in supplementary material, or signposted for the reader if it exists later in the manuscript. It needs to include dosing parameters and specification of any rehabilitation beyond simply practicing functional tasks. 

Our response ⇒ According to the suggestion, we gave a little more detailed account of the common clinical pathway as follows: 

“The common three-week program composed of the followings: The first post-surgical week : exercise for knee joint range of motion (ROM), muscle training for gluteus maximus and gluteus medius, sit-up and stand-up training; the second week: muscle training for quadriceps T-cane walking; the third week: staircase training, daily activity practice, and instructions toward discharge.” 

There is no indication to when recruitment or data collection started, how long data were collected for, or how much missing data there were. I would expect these to be included for completeness. Why was the long-term follow-up of patient difficult? Were patients recruited affected by the COVID pandemic, did they have to self-isolate / were they deconditioned prior to surgery? 

Our response ⇒ We described the period of recruitment as “between July 2013 and February 2018”. The long-term follow-up of patients after surgery was generally difficult because most cases tended to be followed at a local orthopedic clinic. COVID pandemic had nothing to do with the situation. 

I would to see how many eligible participants did not consent to be included in the study for completeness. L99 – the exclusion of co-morbid neurological sensorimotor dysfunction – this I think needs to be more specific e.g. acquired focal neurologic, or chronic deteriorating neurologic condition affecting individuals’ ability to walk. 

L100 – exclusion criterion 2) I think could be made clearer, and criterion 3) would be simpler described as cognitive impairment. 

Our response ⇒ According to the suggestion we modified the description of the exclusion criteria as “The exclusion criteria were 1) presence of motor paralysis or other neurological dysfunctions such as stroke, DM peripheral neuropathy, unrelated to KOA, 2) obvious motion pain or restricted joint movements in locations other than the knee, which would hinder walking or the sit-to-stand motion, and 3) cognitive impairment”.

L115 – how were the healthy volunteers recruited? I would like to see more detail about the healthy participant inclusion criteria 1) and 2) for completeness.

Our response ⇒ We added the following descriptions “We recruited 120 apparently healthy volunteers ….. mostly from elderly clubs registered in a municipal health promotion office”

L123 – the research design sentence could be made more succinct.

Our response ⇒ Following the suggestion we made the description succinct as “This study corresponded to a prospective cohort study for establishing RIs of knee motor functions specific to KOA patients”

Measurements 

I would expect to see the K-L classification referenced please. 

L136 – while the same chair was used for TUG (in all institutions?), its height and whether it had arms would be useful information here for completeness. What was the go signal - Audible? Visual? 

Our response ⇒ We thanks for pointing out the problem. We added the following two references regarding the K-L classification: Kellgren JH, Lawrence JS. Radiological assessment of osteo-arthrosis. Am Rheum Dis 1957; 16:494-502. and Kessler S, Guenther KP, Puhl W. Scoring prevalence and severity in gonarthritis: the suitability of the Kellgren & Lawrence scale. Clin Rheumatol 1998; 17:205-209. 

For the method of TUG test performed using a common protocol, we added the following descriptions in the Methods: “As a common specification, the height of the chair seat without arm-rest was set to 40−45cm. On the vocal “go” signal, the examinee got up from the chair,….” 

L141 – as per above, I am not convinced that Shumway-Cook specified “maximum” tempo in TUG. The MWS excluded acceleration and deceleration phases which is well explained. Like TUG, the go signal for MWS specifics are unclear and should be included, I think. 

Our response ⇒ As mentioned above, we adopted TUG at the maximum tempo by citing the paper of Shumway-Cook in addition to TUG at comfortable pace. This is because both methods are clinically in use in Japan. However, we feel that test results of TUG at the maximum tempo (TUGmax) are more reproducible. For the same reason, in measuring walking speed, we regard that MWS has a higher reproducibility, and thus we adopted MWS in this study.

There is lack of detail within the functional measurement methods section to both justify the approach, allow repeatability, and defend any validity risks. An inclusion of this detail would be beneficial and strengthen the paper. For example, there is no allusion to the reliability or validity of the hand-held dynamometer technique and simply referencing the previous paper (Katoh and Yamasaki, 2009) is not enough. Can the authors account for the quality of the data they used? how many different researchers undertook the measurements? For the muscle testing it was unclear to me whether the ankle was fixed passively to a wall as shown in the Katoh paper, or did the researcher fix the tibial segment manually? A figure might help here. What verbal encouragement was offered to the participants? Was it standardised? What is the justification of taking the mean of 2 (and not more) knee extension muscle contraction attempts? Can the authors account for a first trial response in their data (Allum et al., 2011)? L169 – knee flexion strength is measured in the same manner as knee extension but there is not an allusion to the mean of two attempts for knee flexion strength. I would encourage consistency and clarity in the document. I am unconvinced that the torque calculations could be repeated as it is unclear where the measurement of the dynamometer was taken with reference to the knee joint – i.e. was it the top, bottom, centre-line, other part of the dynamometer?

Our response ⇒ According to the suggestion, we added references on the reproducibility of muscle strength measurement reported by Katoh and Yamasaki (2009), and its validity by Katoh and Yamasaki (2011). With respect to between-institution variability, we believe it is negligible due to generally small between-examiner variability of measurements as reported in those papers. The reason for having measured twice was a higher variability in elderly females as reported by Katoh (2010).

As per advice, we made a new figure with photos (Figure 1) to demonstrate actual procedures to perform the measurements. The position in attaching the HHD sensor was described as follows: “The lower leg length was measured as a distance between the distal end of the lower leg (where the center of the HHD sensor was placed) and the lateral cleft of the knee joint”. More detailed descriptions were also given as legends to the Figure.

The method for measuring knee ROM is extremely thin and should not rely on a reader having to read the reference supplied. It needs to be completely rewritten and expanded for completeness. There is no allusion to what landmarks the femur and tibial segments are designated by nor where the approximation of the knee joint centre is. There is also no allusion in the manuscript to reference positions i.e. is a straight leg 180 degrees extension or 0 degrees flexion? I would expect this detail to be at least included in the supplementary material.

Our response ⇒ We are sorry for the insufficient descriptions of the measurement procedure for ROM. We added description as follows for reproducibility of the measurements: “The angle of the long axes of the tibia (on a line connecting greater trochanter and lateral epicondyle of femur) and the femur (on a line connecting fibula head and lateral malleolus) was measured in units of 5 degrees.”

Statistical Methods

I would like to see some justification about why the candidate factors were selected either by previous work, the authors’ experience, a logical argument, or other in the regression model. The factors selected seem sensible, but why these and not others? As it stands, I am not convinced why e.g. chronicity of OA, smoking, educational level, or living alone were not included as factors. The justification for selecting a rp≥0.2 simply because it is midway between the subjective small and medium correlation effect sizes offered by Cohen (Cohen, 1992) is by itself a weak justification in my opinion; I would prefer to see more justification of effect based on the context of the functional measures undertaken in this paper not just on a reported, yet essentially arbitrary, selection of numbers. 

Our response ⇒ Regarding the selection of explanatory variables, we included all available variables in the regression model. However, we did not record other variables mentions. We wrote about this situation as follows: “The following parameters that we could obtain from all subjects were evaluated as explanatory variables: sex, age, BMI,…”.

For the use of |rp|≥0.2 as our threshold of “effect size”, we simply regarded Cohen’s 0.1 as too weak, and 0.3 as too conservative, and thus arbitrarily took the value in between. However, because most readers are used to P-values in judging significance, and because of the request of Reviewer-1, we added P-value in the regression table to demonstrate the level significance of rp values. At the same time, we modified our descriptions on the effect size as follows:

“In addition to expressing the statistical significance of each standardized partial regression coefficient (rp) in P value, we marked its effect size (practical significance) by use of a threshold of |rp|≥0.2, which corresponds to a midpoint effect size of Cohen’s small (0.1) and medium (0.3) for correlation coefficient [17]” 

The rest of the statistical processing reported appears to be sound and referenced appropriately. I have one query as to why the 2 specific methods of transformation for MRA and SDR calculations were deployed for the TUG and strength data? In calculating the RIs, why was a Box-Cox transformation undertaken for all data without reporting on its normality distribution first? If there are any weaknesses or controversies in the statistical approaches used, then I would like to see them discussed, or at least acknowledged, in the discussion section. 

Our response ⇒ We thank you for raising the important issue of when to transform response and/or explanatory variables before applying multiple regression analysis (MRA) or ANOVA (for computing SDR). There is no consensus criterion on “when to” and “how strictly to” transform values to Gaussian. General practice is to use Box-Cox transformation by selecting appropriate power (‘p’-value). When values are highly skewed, logarithmic transformation (p=0) is commonly used, and when moderately skewed, square-root (p=0.5) or cubic root (p=0.33) transformation are commonly used. However, a minor difference in the selection of power does not cause much difference in MRA results [Please refer to the Appendix at the last two pages of this document]. 

On the other, when we need to determine the central 95% interval strictly by use of parametric method, we must strictly transform value into Gaussian to use mean±1.96SD as the RI after reverse-transformation. This is the reason why the power-value (p) strictly predicted for deriving RIs as shown in Suppl Fig. 1 differs from the power (p=0.0 or 0.5) empirically used in performing MRA and ANOVA. 

If strict prediction of p to attain Gaussian transformation is required for each variable, it should be impossible to perform MRA or ANOVA for most researchers. 

The ethics paragraph is clearly written and welcome, it could be moved to the top of the methods section. Suggest change “We provided participants with a comprehensible document explaining this study…” (L234) to “We provided a written explanation of this study in language that was understandable to participants… “.

Our response ⇒ We followed the advice and moved the section upwards and rephrasing the description.

Results

L248 – I would like to know if the healthy control group were statistically different to the patient participants based on the observed demographics. 

Our response ⇒ We followed the suggestion and prepared a new Table (Table 1) to show demographic comparison between patients and healthy subjects, including results of statistical testing for between-group differences. We added the following description in the Results: “Differences between healthy subjects and KOA patients were tested by Mann-Whiney test for numerical variables and chi-square test for categorical variables. Although actual differences in sex and age distributions were small, BMI was notably higher and proportions of those who exercise regularly were much smaller among KOA patients.” 

Notwithstanding the criticisms already noted, Table 1 and 2 are very clear and their captions thorough.

Our response ⇒ We are pleased to know that the tables are accepted in the current form.

Figure 1 & 2 are a nice visual representation of these data. I note the data are presented transformed and wonder if the non-transformed data might be more meaningful to the reader and justify the use of the median and the inter-quartile range. Not sure why the ROM data are in histograms, the x-axis is already accounting for discreteness in the data.

Our response ⇒ Values for TUGs and MWS are highly skewed with more tailing of values to the higher side. Therefore, without transformation, the data points in the lower side are packed together to identify individual points. In showing median and IQR, we found there was mistake in drawing the statistics, and corrected them in the revised figure.

As for our selection of histogram for showing the distribution of ROM data, the values are discrete in nature (measured in every 5 degree angle) and thus it was not appropriate to use a 1D scatter plot that resulted in vertical stacking of dots at each discrete point of observation.

L370 – “The RIs derived are listed in seven blocks in Table 3. Please note that RIs were derived by a parametric method for motor function tests except for extension and flexor ROMs, for which nonparametric methods were used.” I may have missed it, but the non-parametric version is opaque in the methods section.

Our response ⇒We are sorry about the omission of the method used in deriving the RI non-parametrically. We added the following explanation: “When test results of a given parameter failed in the Gaussian transformation by the Box-Cox method or when test results took discrete values, we applied nonparametric method for determining the central 95% range of test results after sorting them, and took the range of 2.5 and 97.5 percentiles as the limits of the RI.”

Overall, I found the results section too long and suggest some editing to lighten the load on the reader.

Our response ⇒ Following the suggestion, we reduced texts in the Results by 19 lines, although we added 9 lines of new texts for improved legibility 

Discussion

The first 2 paragraphs outlining the background is too long. I think it should simply remind the reader of the objectives, a brief justification of why this is important and/or unique, and perhaps the main findings.

L413 – “Factors associated with presurgical motor functions” section. This alludes to TUG but not which phenotype. Overall this section merely repeats the results which are not particularly compelling and offers no discussion point. As it is, I would remove. 

Our response ⇒ We realized that distinction of TUG was omitted. We amended the problem in the subsection.

In contrast L425 section offers more discussion, but I would remove repeating results with no discussion point.

Our response ⇒ Unfortunately, we found it difficult to remove the discussion on “Factors associated with post-surgical motor functions” with need of citing related studies. We rather removed all the subheading within the Discussion to reduce the text volume, and also trimmed some of texts to make the paragraphs a little more concise.

L437 section alluding to healthy data – this should be a major discussion point, but as outlined above should reference the background where a defence of the need to present data of typical patient populations peri-operatively is required in order to quickly recognise outliers who might require more tailored management following their surgery.

Our response ⇒ Thanks for highlighting the important message of our study written in that paragraph of the Discussion. We modified our description to emphasize our point of “the irrelevance of healthy subjects’ knee function data for our purpose, and the need for the RIs specific for KOA patients in the management of peri-surgical rehabilitation”. 

L446 – “Final decision on partitioning”, and L475 “Reliability of the parametric method ..”section, again these sections could do with some editing and more discussion. It is difficult to clearly map the authors’ objectives for the paper with the results and discussion sections –there were too many surprises in these two sections I was not prepared for making it difficult to read. 

Our response ⇒ We recognized that two sections were difficult to read through, and thus amended the descriptions and trimmed some texts to make our point clearer. 

L497 – “Clinical utility of RIs …” section. Sadly, this is the last, smallest, and rather meagre discussion section. Here I was desperate to see the authors discuss the RIs data. Instead this section lacks anything except that the RIs have been reported. This is a major omission in my opinion and completely under sells the main point of your work. 

Our response ⇒ We admit this problem clearly as we mentioned earlier (the 2nd page of this document), we fully rewrote this block to made it clear that this study originated from physical therapists who are in charge of rehabilitation of KOA patients undergoing knee arthroplasty. We also emphasized the importance of reference intervals of knee motor function tests for improved/objective implementation of peri-surgical rehabilitation, that are supposed to contribute to and predict the long-term success of the surgery, by citing relevant references.

I would like to see much more discussion of the data and the fact that the authors could only follow-up patients 14 days post-surgery due to either lack of resources or because follow up care was too complex. This is alluded to in the limitations section, but why was this design undertaken if this limitation was known a priori?? The conclusion lacks any examples of how the RIs, despite their limitations, could be used objectively in practice by clinicians and researchers.

Our response ⇒ As we wrote at the beginning of our responses, this study was conducted by a team of physical therapists in charge of peri-surgical rehabilitation of KOA patients, who cover the period of before and up to three weeks after the surgery. Therefore, it is difficult to monitor knee motor functions long enough to determine the effectiveness of knee arthroplasty, which was out of our control. 

Finally, there were some trivial edits outlined below to take or leave.

Our response ⇒ We appreciate a great deal for all the kind editing below of our texts throughout the manuscript for improvement. We amended all of them except for the ones that were deleted in the process of revision.

L27-28 The description of the ‘objective peri-surgical management’ in the abstract would be better written as ‘crucial for determining the effectiveness of KA surgery’ or ‘crucial part of objective measures’ 

L50 ‘objective peri-surgical management of KOA’ might be better as ‘determining the effectiveness of KA surgery in patients with KOA.’ 

L55 ‘KOA features degeneration and attrition of joint structure that result in ossification of the cartilage and surrounding tissue, which restricts knee movements and eventually leads to pain and gait disturbance.’ might be better ‘KOA features degeneration and attrition of joint structure that result in ossification of the cartilage and surrounding tissue. This restricts knee range of movement causing pain and gait disturbance (expect a reference here). 

L58-60 ‘Conservative therapy is the primary choice, but with the development of deformity in one or both epicondyles, either unicompartmental knee arthroplasty (UKA) or total knee arthroplasty (TKA) has been the therapeutic regimen of choice.’ Might be better ‘Conservative management is the primary treatment for KOA. However, a TKA / UKA is suggested if the patient fails conservative management’ - plus other clinical indictors previously discussed. 

L63 ‘pre-operative’ would be better ‘pre-operative status’ 

L85 ‘With this background’ could be changed to ‘considering this’

L130 ‘A query was made on the status of regular exercise, and we regarded it positive when 30 minutes or more exercise was performed twice weekly or more.’ Would be better written as ‘Regular exercise was defined as exercise performed a minimum of twice weekly for 30 minutes or more’ 

L135 * L143 probably not use the acronyms in these titles and use full measurement names 

L146 might read better as ‘signal’ rather than ‘sign’ 

L179 I think ‘a’ is missing before ‘multiple regression’, or alternatively plularise 

L190 ‘As a factor for partitioning values in deriving RIs, we considered sex, age, BMI, and surgical mode as candidates.’ Might read better as ‘factors which may influence the RI’s include sex, age, BMI, and surgical mode.’

L241 regular exercise description is in the methods and is duplicated here, consider removing

L305-306 ‘It is obvious that muscle strength is generally lower in the women’ suggest change to ‘extensor muscle strength post-operatively is reduced in females’ 

L345 suggest change from ‘larger’ to ‘greater’ 

L350 suggest change ‘Because’ to ‘As’

L370 suggested change from ‘Please note that RIs were derived by a parametric method for motor function tests except for extension and flexor ROMs, for which nonparametric methods were used. To ‘It is important to note that RIs were derived by a parametric method for motor function tests except for extension and flexor ROMs where nonparametric methods were used instead.’ 

L394 suggested change from ‘and thus they take a long time to reach a favorable level’ to ‘and thus they take a long time to reach their optimum’ 

L394-395 change from ‘Therefore, it is important to detect patients who are lagging behind in the recovery process.’ to ‘Therefore, it is important to detect when patients are not aligning with expected recovering profiles in a timely manner.’ 

L398-402 is a long sentence to parse. Suggest breaking up. 

L404 suggest change ‘ideal’ to ‘important’ 

L430 ‘was’ change to ‘were’

L438 and 441 change ‘suppressed’ to ‘reduced’

L447 suggested change from ‘In judging the need for partitioning of the test results, we relied on SDR.’ ‘we relied on SDR to judge the need for partitioning of the test results.’ 

L455 suggest change ‘but’ to new sentence ‘However,’ 

L460 suggested change from ‘There was a similar issue for’ to ‘There were’ and pluarise

L487 suggest change ‘obviously’ to ‘clearly’

L494 suggested change from ‘again’ to ‘furthermore or nevertheless’ 

L507 ‘data size’ suggested change to ‘the sample size’

L509-510 ‘For the long-term management of KOA patients who undergo knee arthroplasty, lower-limb motor function should be monitored for many months’ suggested change ‘As part of the long-term management of patients who undergo knee arthroplasty, lower-limb motor function scores should be monitored.’ Also suggest adding in why this would be beneficial would strengthen the manuscript. 

L500-501 ‘To the best of our knowledge, there has been no such a report in the past.’ Suggest to remove as repeated in the conclusion 

*6. PLOS authors have the option to publish the peer review history of their article (what does this mean?). If published, this will include your full peer review and any attached files. 

Yes

*Confidential to Editor 

I have no competing interests.

References 

Allum, J.H., Tang, K.S., Carpenter, M.G., Oude Nijhuis, L.B., and Bloem, B.R. (2011). Review of first trial responses in balance control: influence of vestibular loss and Parkinson's disease. Hum Mov Sci 30(2), 279-295.

Bohannon, R.W. (2006). Reference values for the timed up and go test: a descriptive meta-analysis. J. Geriatr. Phys. Ther. 29(2), 64-68.

Cohen, J. (1992). A power primer. Psychol. Bull. 112(1), 155-159.

Kamide, N., Takahashi, K., and Shiba, Y. (2011). Reference values for the Timed Up and Go test in healthy Japanese elderly people: determination using the methodology of meta-analysis. Geriatr Gerontol Int 11(4), 445-451.

Katoh, M., and Yamasaki, H. (2009). Comparison of Reliability of Isometric Leg Muscle Strength Measurements Made Using a Hand-Held Dynamometer with and without a Restraining Belt. J Phys Ther Sci 21(1), 37-42.

Podsiadlo, D., and Richardson, S. (1991). The timed "Up & Go": a test of basic functional mobility for frail elderly persons. J. Am. Geriatr. Soc. 39(2), 142-148.

Sekiya, N., Nagasaki, H., Ito, H., and Furuna, T. (1997). Optimal walking in terms of variability in step length. J. Orthop. Sports Phys. Ther. 26(5), 266-272.

Shumway-Cook, A., Brauer, S., and Woollacott, M. (2000). Predicting the probability for falls in community-dwelling older adults using the Timed Up & Go Test. Phys. Ther. 80(9), 896-903.

---

## [Decision Letter · Decision Letter 1]

27 Jan 2021

PONE-D-20-18248R1

Determination of reference intervals for knee motor functions specific to patients undergoing knee arthroplasty

PLOS ONE

Dear Dr. Ichihara,

Thank you for submitting your manuscript to PLOS ONE. After careful consideration, we feel that it has merit but does not fully meet PLOS ONE’s publication criteria as it currently stands. Therefore, we invite you to submit a revised version of the manuscript that addresses the points raised during the review process.

ACADEMIC EDITOR:

The reviewers feed back positively on the changes that have been made to improve the manuscript but still raise a number of issues. Please address the reviewers further comments to enable greater clarity within the manuscript.

We look forward to receiving your revised manuscript.

Kind regards,

Alison Rushton

Academic Editor

PLOS ONE

Reviewers' comments:

Reviewer's Responses to Questions

**Comments to the Author**

1. If the authors have adequately addressed your comments raised in a previous round of review and you feel that this manuscript is now acceptable for publication, you may indicate that here to bypass the “Comments to the Author” section, enter your conflict of interest statement in the “Confidential to Editor” section, and submit your "Accept" recommendation.

Reviewer #1: All comments have been addressed

Reviewer #2: (No Response)

Reviewer #3: (No Response)

2. Is the manuscript technically sound, and do the data support the conclusions?

Reviewer #1: Yes

Reviewer #2: Partly

Reviewer #3: Partly

3. Has the statistical analysis been performed appropriately and rigorously? 

Reviewer #1: Yes

Reviewer #2: No

Reviewer #3: Yes

4. Have the authors made all data underlying the findings in their manuscript fully available?

Reviewer #1: Yes

Reviewer #2: Yes

Reviewer #3: Yes

5. Is the manuscript presented in an intelligible fashion and written in standard English?

Reviewer #1: Yes

Reviewer #2: Yes

Reviewer #3: Yes

6. Review Comments to the Author

Reviewer #1: 

The comment from previous review have been addressed, but some minor issues have arisen in newly added sections.

They are listed below with line numbers refering to text with tracked changes.

Line 226: misspelled name „Dbson” instead of Dobson

Lines 258-260: description of tibia and femur axis seem to be swapped; description for tibia fits to femur and reversely description for femur fits to tibia.

Table 1, row about sex: plural “men” should be written instead of singular “man”.

Lines 396-400: First, an information about extensor strength reduction in women is given. Next, it is repeated for both sexes, which is true, but there is no need to state it twice.

Line 516: strictly speaking, no data from Table 4 directly prove that distribution was not Gaussian before transform.

Reviewer #2: 

Although the authors made most of the suggestions, the objectives, results presented and discussions are still not clear for the reader to follow a clinical reason.

I suggest that the authors write the clinical meaning for each result found in the discussion.

Reviewer #3: 

The authors have improved the manuscript based on all reviewers’ comments. The rationale is clearer and the statistical techniques are better described now

Overall, I still find the manuscript difficult to read and I contend it could be made clearer and simpler with better mapping of the stated objectives to the results.

I think the authors want to say that previous reporting of RIs in predominantly healthy subjects are flawed statistically and offer methods as an alternative approach which is a valid purpose. Reporting RIs in healthy subjects using the alternative statistical technique might be the sole purpose of a separate paper. They then suggest that the reporting of specific motor functions (TUG, max walking velocity, leg ROM, & specified lower limb strength) is relevant. I am still to be convinced by the authors’ manuscript that they recognise that these measures are examples of proxies of physical functional constructs and that there exist other measures – so why these specific ones and not others, and why measured this way?? Again, this debate might provide a separate manuscript worthy of publication.

Regardless of this criticism, they go on to provide RI data for these measures which is in itself sensible and useful for discriminative use in clinical practice or research given the statistical flaws described in previous work. They content that these data are useful in clinical Japanese practice as a means to determine if a patient is starting from a deranged functional profile pre-surgery, or is deviating from a normal recovery trajectory after surgery based on the functional measures selected.

But, this reader is not convinced the authors have justified that their sample represents a typical patient sample. This is mainly because they openly admit they were limited in their design to determine whether their sample reached a pre-determined threshold of functional recovery longitudinally i.e. whether the longer-term outcome of their surgery was successful. We simply do not know if this is the case, and therefore there is doubt their RIs represent a compelling acceptable range. This needs to be better acknowledged in my opinion.

All in all, I think the authors have a made a decent attempt effort to incorporate and/or rebut the 3 reviewers’ comments. While the manuscript is improved, it still requires some revision. It could benefit from breaking into clear sections or even be incorporated into two manuscripts in my opinion.

Specific points and minor Edits (line numbers refer to tracked-changed manuscript version):

L82 onwards - Explain what the differences between the different surgical types in the introduction after L82.

L43 -Change ‘as controls’ to ‘a control group of 120 healthy elderly volunteers’

L72 - Change ‘by Yoshimura et al.’ to ‘Yoshimura et al. (2011) reported in a large cohort study the prevalence of knee osteoarthritis (KOA) is as high as 42.6% in men and 62.4% in women over 40 years old across Japan.

L85-88 - Change ‘For proper management and smooth recovery through rehabilitation, it is important to monitor knee motor functions [4] such as timed-up-and-go (TUG) test, maximum walking speed (MWS), knee muscle strength, and knee range of motion (ROM).’ To ‘It is important to use objective tests such as timed-up-and-go (TUG) test, maximum walking speed (MWS), knee muscle strength, and knee range of motion (ROM) to monitoring performance post operatively [4].’

L113 - Change objective 1 to ‘to explore sources of variation (SVs) and determine their reference interval (RIs) of objective tests used to measure knee motor function within KOA patients undergoing knee arthroplasty’

L116 - Change objective 2 ‘To be the first study to explore statistical methods for analysing the SVs and determining the RIs.’

L118 -Change objective 3 to ‘To compare knee motor functions before and two-weeks after knee arthroplasty as a predictor of long-term success of KA surgery.’

L124 - Change ‘A total of 583 KOA patients undergoing elective knee arthroplasty were recruited consecutively between July 2013 and February 2018 by use of harmonized study protocol from 13 institutions specialized in knee arthroplasty, which are scattered widely in western Japan.’ To ‘A total of 583 KOA patients undergoing elective knee arthroplasty were recruited from 13 institutions across Western Japan that specialized in knee arthroplasty between July 2013 and February 2018’.

L140-149 – The objectives are clearer now. But similar to previous criticisms of a lack of a clear mapping of objectives to the data collected - I do not see why a group of healthy subjects was recruited with these objectives in mind – the justification was not made clear to me in the methods section either. I had to read ahead to L395 in the results for a first mention of how the healthy data was used and why.

L161 – Define DM in full if using; but surely peripheral neuropathy of any aetiology would be the exclusion criteria not only diabetes?

L165 – “… enrolled were 545 subjects.” Suggest change to “…545 subjects were enrolled.”

L173 – Change “followings” to “following”

L211 – Change “… KOA patients …” to “… KOA patient …”

L214- Suggest change “… on a chair …” to “… on an armless chair with seat-height set between 40 and 40cm high …” and delete the next sentence.

L258-260 – I think the descriptors of the tibial and femoral segments need changing around.

L275 – “… in P value …” is better as “ … as a P value …”

L277 – pluralise “… coefficient …” I think

L317 – suggest change “… we applied nonparametric …” to “… we applied a nonparametric …”

L339 – “… Mann-Whiney…” should be “… Mann-Whitney …”

L340 – I think pluralise the word “test” when it appears in this sentence, and these analyses should be in the methods section.

L340-342 - Allusion to, or interpretations of, “small” or any statistical differences should be included in the discussion section – the results should merely report the results I think

L408 – Table 3, there is no adjacent mention of Table 3 in the text here, I had to look to L424 in the next section to find it, Table 3 needs better placement in the text

L506 – suggest change “… proper …” to “ … optimal …” or similar

L580 – I am not sure you need “on the other hand ….” At the start of the sentence here

7. PLOS authors have the option to publish the peer review history of their article (what does this mean?). If published, this will include your full peer review and any attached files.

Reviewer #1: No

Reviewer #2: No

Reviewer #3: **Yes: **Gareth D. Jones

---

## [Author Response · Author response to Decision Letter 1]

16 Feb 2021

Reviewer #1

Our response ⇒ We greatly appreciate your review of our manuscript and the invaluable comments provided to improve it. We addressed each issue and revised the manuscript accordingly as described below.

1. Line 226: misspelled name „Dbson” instead of Dobson

2. Lines 258-260: description of tibia and femur axis seem to be swapped; description for tibia fits to femur and reversely description for femur fits to tibia.

3. Table 1, row about sex: plural “men” should be written instead of singular “man”.

4. Lines 396-400: First, an information about extensor strength reduction in women is given. Next, it is repeated for both sexes, which is true, but there is no need to state it twice. 

Our response ⇒We corrected the errors as suggested.

5. Line 516: strictly speaking, no data from Table 4 directly prove that distribution was not Gaussian before transform.

Our response ⇒ We thank you for pointing out this mistake. We removed the following phrase from Table 4: “the distribution of TUG values is not Gaussian as is evident from our Table. 4 and Suppl Fig. 1”

Reviewer #2

- Although the authors made most of the suggestions, the objectives, results presented and discussions are still not clear for the reader to follow a clinical reason.

I suggest that the authors write the clinical meaning for each result found in the discussion.

Our response ⇒ We are grateful for your review of our manuscript again. We realized the problem of describing our objectives of this study comprehensibly in the Introduction and the lack of our statement of clinical implications of this study in the Discussion. We concisely described the objectives as follow:

“With these backgrounds, we conducted a multicenter study to evaluate sources of variation of knee motor functions and to establish their RIs specific for KOA patients undergoing KA. 

We aimed at using the quantitative information for objective implementation of peri-surgical rehabilitation and for screening patients who are deviated from the normal course of recovery.”

We added the clinical implication of this study as follows at the end of the Discussion:

“In summary, a clinical implication of this study is as follows. Since we only enrolled patients who followed the normal course of recovery, the quantitative information of RIs and their factors of variations obtained in this study will facilitate peri-surgical rehabilitation customized to patients according to sex, age, and surgical mode. Furthermore, the subgroup specific RIs will be useful in detecting patients who are deviated from the normal course of recovery.”

 

Reviewer #3

The authors have improved the manuscript based on all reviewers’ comments. The rationale is clearer and the statistical techniques are better described now

Overall, I still find the manuscript difficult to read and I contend it could be made clearer and simpler with better mapping of the stated objectives to the results.

Our response ⇒ We are grateful for your time and efforts to review our manuscript again in great detail to give us a chance to improve it.

First of all, we realized that the following three descriptions of our objectives in the Introduction were still incomprehensible and misleading and thus caused further confusion. 

“With these backgrounds, we conducted this study as a team of physical therapists to provide optimal peri-surgical rehabilitation to KOA patients undergoing knee arthroplasty in collaboration with orthopedicians. Our specific objectives were:

(1) For objective use of knee motor function tests in the rehabilitation, to explore their sources of variation (SVs) and to determine their reference interval (RIs), which are specific to KOA patients undergoing knee arthroplasty. 

(2) To apply up-to-date statistical methods in analyzing the SVs and in determining RIs, which have not been used in the field of rehabilitation medicine in the past,

(3) To compare knee motor functions before and two-weeks after knee arthroplasty, which is of relevance in predicting a long-term success of the surgery. “

We think that this problem occurred due to an insufficient description of the role of physical therapists in the post-surgical rehabilitation of patients undergoing knee arthroplasty (KA). Therefore, we added the following description of how the peri-surgical rehabilitation is generally performed and how the knee function tests are being used by adding a new Figure 1.

“In the clinical management of KOA patients undergoing knee arthroplasty, physical therapists take an important role in providing peri-surgical rehabilitation in close collaboration with orthopedic surgeons. The scheme of rehabilitation according to a common clinical pathway for knee arthroplasty is as shown in Fig. 1. The functional status of patients is usually assessed by performing knee motor function tests of a safe variety, such as timed-up-and-go (TUG) test, maximum walking speed (MWS), knee muscle strength, and knee range of motion (ROM) [4]. 

These test results are often referred to in deciding a patient’s discharge date. To read the results objectively, it is essential to understand their sources of variation and to have a reliable reference interval (RI) for each parameter.”

Therefore, we adopted only the first objective and deleted other two because of their misleading nature and irrelevancy for this study. 

The first objective is now re-written as follows at the end of the Introduction:

“On the basis of this background, we conducted a multicenter study to evaluate sources of variation of knee motor functions and to establish specific RIs for KOA patients undergoing KA. We aimed to use quantitative information for objective implementation of peri-surgical rehabilitation and for screening of patients who deviate from the normal course of recovery.”

As for objective (2) of applying up-to-date statistical methods, we now regard it not so important and thus we toned down this point and substantially reduced discussion on the past “normative value” studies.

For objective (3) of comparing knee functions before and after KA, we found the description was misleading because immediate post-surgical test results from our study cannot be used in predicting long-term patient outcome, although there are some studies that claimed the importance of immediate postsurgical knee functions in relation to long term outcome.

I think the authors want to say that previous reporting of RIs in predominantly healthy subjects are flawed statistically and offer methods as an alternative approach which is a valid purpose. Reporting RIs in healthy subjects using the alternative statistical technique might be the sole purpose of a separate paper. 

Our response ⇒ As we just mentioned above, we found it irrelevant to cite past studies of knee motor functions based on healthy elderly subjects, regardless of the statistical methods used. Therefore, we deleted some of those description in the Introduction and the Discussion. Thus, we have no plans to write a companion paper to this one because the two populations (healthy vs. KOA) are too mismatched to make any worthy comparison.

They then suggest that the reporting of specific motor functions (TUG, max walking velocity, leg ROM, & specified lower limb strength) is relevant. 

I am still to be convinced by the authors’ manuscript that they recognise that these measures are examples of proxies of physical functional constructs and that there exist other measures – so why these specific ones and not others, and why measured this way?? 

Again, this debate might provide a separate manuscript worthy of publication.

Our response ⇒ We understand that there are many other measures of knee motor functions, such as the 6-minute walking distance and a stair climb test. However, during the immediate post-surgical period that physical therapists are involved in, these tests are generally not performed for the sake of safety. Unfortunately, a second study targeting other physical function tests is beyond the scope of our routine clinical practice. 

Regardless of this criticism, they go on to provide RI data for these measures which is in itself sensible and useful for discriminative use in clinical practice or research given the statistical flaws described in previous work. 

Our response ⇒ We toned down the issue of statistical flaws in the past normative value studies in the Discussion because of the irrelevance of the issue based on the main objectives of this study.

They content that these data are useful in clinical Japanese practice as a means to determine if a patient is starting from a deranged functional profile pre-surgery, or is deviating from a normal recovery trajectory after surgery based on the functional measures selected.

But, this reader is not convinced the authors have justified that their sample represents a typical patient sample. 

This is mainly because they openly admit they were limited in their design to determine whether their sample reached a pre-determined threshold of functional recovery longitudinally i.e. whether the longer-term outcome of their surgery was successful. 

We simply do not know if this is the case, and therefore there is doubt their RIs represent a compelling acceptable range. This needs to be better acknowledged in my opinion.

Our response ⇒ We again apologize for having cited misleading papers in the Discussion, which reported the importance of short-term post-surgical motor functions in predicting the long-term recovery of patients. We admit that we cannot say anything about the long-term success of the surgery.

As for the implication of the RIs we derived, we had missed important descriptions on how we enrolled KOA patients in the Methods. We just included patients who experienced a normal recovery course and were discharged within 3 to 4 weeks after KA. Therefore, our RIs can be referred to in judging whether a given patient is in a normal recovery course for scheduling discharge. We also need to emphasize that the clinical pathway we adopted in this study is widely used in Japan, and thus the results of this study are relevant in the clinical practise of peri-surgical rehabilitation of KA patients in Japan.

Since we clarified the sources of variation of the test results, an additional clinical implication of this study is that it was made possible to interpret test results more appropriately in considerations of sex, age, and surgical mode.

In the revised manuscript, we added these points as a clinical implication of this study in the last section of the Discussion:

“In summary, the clinical implication of this study is as follows. As we only enrolled patients who followed a normal course of recovery, the quantitative information of RIs and their factors of variation obtained in this study will facilitate peri-surgical rehabilitation customized to patients according to sex, age, and surgical mode. Furthermore, the subgroup-specific RIs will be useful in detecting patients who deviate from the normal course of recovery.”

All in all, I think the authors have a made a decent attempt effort to incorporate and/or rebut the 3 reviewers’ comments. While the manuscript is improved, it still requires some revision. 

It could benefit from breaking into clear sections or even be incorporated into two manuscripts in my opinion.

Our response ⇒ We hope we have clarified our objective and clinical implications of this study as described above in responding to your invaluable suggestions one by one. We also hope that it will not be necessary to write other related manuscripts, which seems not possible from the objective of this study.

Specific points and minor Edits (line numbers refer to tracked-changed manuscript version):

L82 onwards - Explain what the differences between the different surgical types in the introduction after L82.

Our response ⇒ We added the following description of the surgical modes: 

“Either unicompartmental knee arthroplasty (UKA) or total knee arthroplasty (TKA) has been the therapeutic regimen of choice. The former is a less invasive technique that only replaces a single granular area and preserves the anterior and posterior cruciate ligaments [2,3]. Recently, as an alternative to conventional TKA (C-TKA), a minimally invasive TKA surgery (MIS-TKA) has become popular, which features a shorter skin incision than that with C-TKA.”

L43 -Change ‘as controls’ to ‘a control group of 120 healthy elderly volunteers’

L72 - Change ‘by Yoshimura et al.’ to ‘Yoshimura et al. (2011) reported in a large cohort study the prevalence of knee osteoarthritis (KOA) is as high as 42.6% in men and 62.4% in women over 40 years old across Japan.

L85-88 - Change ‘For proper management and smooth recovery through rehabilitation, it is important to monitor knee motor functions [4] such as timed-up-and-go (TUG) test, maximum walking speed (MWS), knee muscle strength, and knee range of motion (ROM).’ To ‘It is important to use objective tests such as timed-up-and-go (TUG) test, maximum walking speed (MWS), knee muscle strength, and knee range of motion (ROM) to monitoring performance post operatively [4].’

L113 - Change objective 1 to ‘to explore sources of variation (SVs) and determine their reference interval (RIs) of objective tests used to measure knee motor function within KOA patients undergoing knee arthroplasty’

L116 - Change objective 2 ‘To be the first study to explore statistical methods for analysing the SVs and determining the RIs.’

Our response ⇒We revised the wording as suggested.

L118 -Change objective 3 to ‘To compare knee motor functions before and two-weeks after knee arthroplasty as a predictor of long-term success of KA surgery.’

Our response ⇒We deleted the misleading objective 3 as explained above.

L124 - Change ‘A total of 583 KOA patients undergoing elective knee arthroplasty were recruited consecutively between July 2013 and February 2018 by use of harmonized study protocol from 13 institutions specialized in knee arthroplasty, which are scattered widely in western Japan.’ To ‘A total of 583 KOA patients undergoing elective knee arthroplasty were recruited from 13 institutions across Western Japan that specialized in knee arthroplasty between July 2013 and February 2018’.

Our response ⇒We corrected the description as suggested. Please note that in the revised manuscript, we stated that we originally recruited 623 KOA patients, but we excluded 41 subjects who did not follow a normal course of recovery as well as 38 subjects with missing test results. Therefore, after the above revised description, we added the following facts about the enrolled cases.

 “After exclusion of 38 patients with missing motor function test results and 41 patients who had a delay in discharge, a total of 545 patients were regarded as eligible for the subsequent analyses.”

L140-149 – The objectives are clearer now. But similar to previous criticisms of a lack of a clear mapping of objectives to the data collected - I do not see why a group of healthy subjects was recruited with these objectives in mind – the justification was not made clear to me in the methods section either. I had to read ahead to L395 in the results for a first mention of how the healthy data was used and why.

Our response ⇒ As mentioned above, we admitted the irrelevance of knee motor functions of healthy subjects, and we just described past normative value studies briefly. Nevertheless, to show the needs of the RIs specific for KOA patients, we think it necessary to show the test results of age-matched healthy subjects in this study. Therefore, we hope the healthy subjects’ results can be retained in the Results to emphasize the importance of deriving “disease-specific” RIs.

L161 – Define DM in full if using; but surely peripheral neuropathy of any aetiology would be the exclusion criteria not only diabetes?

Our response ⇒ We revised the description of the excluded conditions as follows:

“1) presence of motor paralysis or other neurological dysfunctions such as stroke, lumbar disc herniation, spinal canal stenosis, peripheral neuropathy of diabetes mellitus unrelated to KOA,…”

L165 – “… enrolled were 545 subjects.” Suggest change to “…545 subjects were enrolled.”

L173 – Change “followings” to “following”

L211 – Change “… KOA patients …” to “… KOA patient …”

L214- Suggest change “… on a chair …” to “… on an armless chair with seat-height set between 40 and 40cm high …” and delete the next sentence.

L258-260 – I think the descriptors of the tibial and femoral segments need changing around.

L275 – “… in P value …” is better as “ … as a P value …”

L277 – pluralise “… coefficient …” I think

L317 – suggest change “… we applied nonparametric …” to “… we applied a nonparametric …”

L339 – “… Mann-Whiney…” should be “… Mann-Whitney …”

L340 – I think pluralise the word “test” when it appears in this sentence, and these analyses should be in the methods section.

Our response ⇒We thank you for these suggestions, which we applied to the manuscript.

L340-342 - Allusion to, or interpretations of, “small” or any statistical differences should be included in the discussion section – the results should merely report the results I think

Our response ⇒According to the suggestion, we rephrased the results as follows removing the interpretations: “Statistically significant differences were found for higher age and BMI, and less frequent habit of exercise observed among KOA patients.” 

L408 – Table 3, there is no adjacent mention of Table 3 in the text here, I had to look to L424 in the next section to find it, Table 3 needs better placement in the text

L506 – suggest change “… proper …” to “ … optimal …” or similar

L580 – I am not sure you need “on the other hand ….” At the start of the sentence here

Our response ⇒ We thank you for these suggestions, which we accordingly applied to the manuscript.

---

## [Decision Letter · Decision Letter 2]

4 Mar 2021

PONE-D-20-18248R2

Determination of reference intervals for knee motor functions specific to patients undergoing knee arthroplasty

PLOS ONE

Dear Dr. Ichihara,

Thank you for submitting your manuscript to PLOS ONE. After careful consideration, we feel that it has merit but does not fully meet PLOS ONE’s publication criteria as it currently stands. Therefore, we invite you to submit a revised version of the manuscript that addresses the points raised during the review process.

ACADEMIC EDITOR:

Thank you for your diligence in addressing the reviewers' comments that I hope you agree has contributed to a stronger manuscript. Please address the few remaining issues raised by the reviewers that are detailed below.

We look forward to receiving your revised manuscript.

Kind regards,

Alison Rushton

Academic Editor

PLOS ONE

Journal Requirements:

Reviewers' comments:

Reviewer's Responses to Questions

**Comments to the Author**

1. If the authors have adequately addressed your comments raised in a previous round of review and you feel that this manuscript is now acceptable for publication, you may indicate that here to bypass the “Comments to the Author” section, enter your conflict of interest statement in the “Confidential to Editor” section, and submit your "Accept" recommendation.

Reviewer #1: All comments have been addressed

Reviewer #3: (No Response)

2. Is the manuscript technically sound, and do the data support the conclusions?

Reviewer #1: Yes

Reviewer #3: Yes

3. Has the statistical analysis been performed appropriately and rigorously? 

Reviewer #1: Yes

Reviewer #3: Yes

4. Have the authors made all data underlying the findings in their manuscript fully available?

Reviewer #1: Yes

Reviewer #3: Yes

5. Is the manuscript presented in an intelligible fashion and written in standard English?

Reviewer #1: Yes

Reviewer #3: Yes

6. Review Comments to the Author

Reviewer #1: 

Previous comments has been addressed and manuscript is in general sound.

Some minor changes, listed below, are still needed, but they do not affect the content substantially. Line numbers below refer to the text with highlighted changes.

Lines 355-363: The authors probably meant Fig 3, not 2, here.

Line 435: There should be Fig 4, not 3.

Reviewer #3: 

The authors have made an excellent attempt to consider all the reviewer’s comments and there is no doubt this is an improved manuscript. The manuscript provides RI data for the functional measures in a real set of patients which is sensible and of use to the community for discriminative use in clinical practice or research given the statistical flaws described in previous work. They contend that these data are useful in clinical Japanese practice as a means to determine if a patient is starting from a deranged functional profile pre-surgery, or is deviating from a normal recovery trajectory after surgery based on the functional measures selected.

Overall, I still find the manuscript difficult to read. Suggest to include in line 88 in the introduction what the sources of variation are/might be to prime the reader to the factors explored in the paper. I still think it could be made clearer and simpler with better mapping of the stated objectives to the results – e.g. crafting the manuscript so the objective of identifying factors associated with motor function parameters is clearly stated at the end of the introduction and in the methods, and a clear rationale why healthy subjects were included (again at the end of the intro and in the methods section from line 144), and some allusion to discussing the healthy data in the discussion section.

7. PLOS authors have the option to publish the peer review history of their article (what does this mean?). If published, this will include your full peer review and any attached files.

Reviewer #1: No

Reviewer #3: **Yes: **Gareth D Jones

---

## [Author Response · Author response to Decision Letter 2]

14 Mar 2021

Reviewer #1: 

Previous comments has been addressed and manuscript is in general sound.

Some minor changes, listed below, are still needed, but they do not affect the content substantially. Line numbers below refer to the text with highlighted changes.

Lines 355-363: The authors probably meant Fig 3, not 2, here.

Line 435: There should be Fig 4, not 3.

Our response → We thanks again for the review of our revised manuscript and the provision of the comment on the problem of incorrect numbering of Figures. We corrected them accordingly. 

Reviewer #3: 

The authors have made an excellent attempt to consider all the reviewer’s comments and there is no doubt this is an improved manuscript. The manuscript provides RI data for the functional measures in a real set of patients which is sensible and of use to the community for discriminative use in clinical practice or research given the statistical flaws described in previous work. They contend that these data are useful in clinical Japanese practice as a means to determine if a patient is starting from a deranged functional profile pre-surgery, or is deviating from a normal recovery trajectory after surgery based on the functional measures selected.

Our response → We are grateful for your time and efforts to review our manuscript again and the kind offering to us of additional comments to further improve the manuscript.

Overall, I still find the manuscript difficult to read. Suggest to include in line 88 in the introduction what the sources of variation are/might be to prime the reader to the factors explored in the paper. 

Our response → According to your suggestion, we added the following explanation for the term of “sources of variation” at line 88: “To read the results objectively, it is essential to understand their sources of variation (factors that may alter test results without pathological conditions) and to have a reliable reference interval (RI) for each parameter.”

I still think it could be made clearer and simpler with better mapping of the stated objectives to the results – e.g. crafting the manuscript so the objective of identifying factors associated with motor function parameters is clearly stated at the end of the introduction and in the methods, and a clear rationale why healthy subjects were included (again at the end of the intro and in the methods section from line 144), and some allusion to discussing the healthy data in the discussion section.　

Our response → According to your advice, we also added the meaning of analyzing the sources of variation (SV) as follow near the end of the Introduction: “On the basis of this background, we conducted a multicenter study to evaluate sources of variation (SVs) of knee motor functions and to establish well-stratified RIs in consideration of the SVs for KOA patients undergoing knee arthroplasty.”. 

Furthermore, the objective of stratifying the RIs in consideration of SVs had been mapped in the Results under the subheading of “Assessment of factors to derive RIs specific to subgroups” (just before the subheading of “Determination of RIs”) in describing the SV-based stratification of the RIs for the KOA patients.

Regarding the implication of recruiting healthy subjects and determining their RIs for the knee motor functions tests, we also added the following description at the very end of the Introduction to clarify the point according to your suggestion: “Age matched healthy subjects were also recruited to ascertain the need for disease -specific RIs. We aimed to use these quantitative information for objective implementation of peri-surgical rehabilitation and for screening of patients who deviate from the normal course of recovery.”

For this issue of our deliberate inclusion of age-matched healthy elderly, we also gave additional explanation of including them in the Methods under the sub-heading of “Healthy volunteers” as follows: “As described in the Introduction, we recruited 120 apparently healthy volunteers with comparable ages to make a contrast of their knee motor functions with those of KOA patients undergoing knee arthroplasty. They comprised 36 men and 84 women ……”

As for the need in the Discussion of additional description for the implication of deriving RIs from the healthy individuals, we had already described it as follows at the third paragraph of the Discussion: “In addition to these methodological issues of the past studies targeting healthy subjects, as demonstrated in Fig. 3 and 4, healthy ranges of knee motor function tests were quite different from those of KOA patients both before and after the surgery.…”

---

## [Editor Report · Decision Letter 3]

22 Mar 2021

Determination of reference intervals for knee motor functions specific to patients undergoing knee arthroplasty

PONE-D-20-18248R3

Dear Dr. Ichihara,

We’re pleased to inform you that your manuscript has been judged scientifically suitable for publication and will be formally accepted for publication once it meets all outstanding technical requirements.

Kind regards,

Alison Rushton

Academic Editor

PLOS ONE

Additional Editor Comments (optional):

Thank you for your diligence in addressing the reviewers' comments at each stage. I hope that you agree that it has strengthened your manuscript.

---

## [Editor Report · Acceptance letter]

5 Apr 2021

PONE-D-20-18248R3 

Determination of reference intervals for knee motor functions specific to patients undergoing knee arthroplasty 

Dear Dr. Ichihara:

I'm pleased to inform you that your manuscript has been deemed suitable for publication in PLOS ONE. Congratulations! Your manuscript is now with our production department. 

Kind regards, 

on behalf of

Professor Alison Rushton 

Academic Editor

PLOS ONE